# MMFNet: Multi-Scale Frequency Masking Neural Network for Time Series Forecasting

## Abstract

Long-term Time Series Forecasting (LTSF) is critical for numerous real-world applications, such as electricity consumption planning, financial forecasting, and disease propagation analysis. Time series, generated from continuous real-world processes sampled at multiple scales, pose significant challenges for LTSF. These challenges arise from the need to capture long-range dependencies between inputs and outputs, driven by the complex temporal dynamics of multi-scale, multi-periodic data. While current time-domain multiscale models effectively capture temporal variations, they often fall short with multi-scale datasets. These models primarily focus on temporal patterns, frequently overlooking critical frequency-specific features, such as harmonics and periodic behaviors, which are better represented in the frequency domain. In this paper, we introduce MMFNet, a novel model designed to enhance long-term multivariate forecasting by leveraging a multi-scale masked frequency decomposition approach. MMFNet captures fine, intermediate, and coarse-grained temporal patterns by converting time series into frequency segments at varying scales while employing a learnable mask to filter out irrelevant components adaptively.

Extensive experimentation with benchmark datasets shows that MMFNet not only addresses the limitations of the existing methods but also consistently achieves good performance. Specifically, MMFNet achieves up to $6.0\%$ reductions in the Mean Squared Error (MSE) compared to state-of-the-art models designed for multivariate forecasting tasks.

## 1 Introduction

Time series forecasting is pivotal in a wide range of domains, such as environmental monitoring (Bhandari et al., 2017), electrical grid management (Zufferey et al., 2017), financial analysis (Sezer et al., 2020), and healthcare (Zeroual et al., 2020). Accurate long-term forecasting is essential for informed decision-making and strategic planning. Traditional methods, such as autoregressive (AR) models (Nassar et al., 2004), exponential smoothing (Hyndman & Athanasopoulos, 2008), and structural time series models (Harvey, 1989), have provided a robust foundation for time series analysis by leveraging historical data to predict future values. However, real-world systems frequently exhibit complex, non-stationary behavior, with time series characterized by intricate patterns such as trends, fluctuations, and cycles. Those complexities pose significant challenges to achieving accurate forecasts (Makridakis et al., 1998; Box et al., 2015).

Long-term Time Series Forecasting (LTSF) has seen significant advancements in recent years, driven by the development of sophisticated models, such as Transformer-based models (Zhou et al., 2021; Wu et al., 2021; Nie et al., 2024) and linear models (Zeng et al., 2023; Xu et al., 2024; Lin et al., 2024). Transformer-based architectures have demonstrated exceptional capacity in capturing complex temporal patterns by effectively modeling long-range dependencies through self-attention mechanisms at the cost of heavy computation workload, particularly when facing large-scale time series data, which significantly limits their practicality in real-time applications. In contrast, the linear models provide a lightweight alternative for real-time forecasting. In particular, FITS demonstrates superior predictive performance across a wide range of scenarios with only $10K$ parameters by utilizing a single-scale frequency domain decomposition method combined with a low-pass filter employing a fixed cutoff frequency (Xu et al., 2024).

Current methods often overlook the multiscale periodic nature of time series data. Time series are generated from continuous real-world processes sampled at various scales. For example, daily data capture hourly fluctuations, while yearly data reflect long-term trends and seasonal cycles. This inherent multi-scale, multi-periodic characteristic presents a significant challenge for model design, as each scale emphasizes distinct temporal dynamics that need to be effectively captured. Centered Kernel Alignment analysis has shown the ability to produce diverse representations across layers is particularly beneficial for tasks requiring the capture of irregular patterns Kornblith et al. (2019). These diverse representations are instrumental in managing variations across scales and periodicities.

Current time-domain multiscale models like TimeMixer (Wang et al., 2024), though effective at capturing temporal variations across resolutions, has several limitations, particularly for datasets with multi-scale and multi-periodic properties. It primarily focuses on temporal patterns, often overlooking critical frequency-specific features such as harmonic or periodic behaviors, which are better captured in the frequency domain. For example, seasonal or cyclic trends are more apparent in frequency representations but can be difficult to disentangle in the time domain. Additionally, time-domain methods are sensitive to noise, as they work directly on raw signals, allowing noise to propagate across scales and obscure meaningful patterns, especially at coarser resolutions. Furthermore, while these methods enhance temporal resolution, they frequently struggle to capture long-term dependencies, as dividing data into scales can result in a loss of the broader context necessary for understanding long-range interactions.

In this paper, we present MMFNet, a novel model designed to enhance LTSF through a multi-scale masked frequency decomposition approach. MMFNet captures fine, intermediate, and coarse-grained patterns in the frequency domain by segmenting the time series at multiple scales. At each scale, MMFNet employs a learnable mask that adaptively filters out irrelevant frequency components based on the segment's spectral characteristics. MMFNet offers two key advantages: (i) the multi-scale frequency decomposition enables MMFNet to effectively capture both short-term fluctuations and broader trends in the data, and (ii) the learnable frequency mask adaptively filters irrelevant frequency components, allowing the model to focus on the most informative signals. These features make MMFNet well-suited to capturing both short-term and long-term dependencies in complex time series, positioning it as an effective solution for various LTSF tasks.

In summary, the contributions of this paper are as follows:

- To our knowledge, MMFNet is the first model that employs multi-scale frequency domain decomposition to capture the dynamic variations in the frequency domain;

- MMFNet introduces a novel learnable masking mechanism that adaptively filters out irrelevant frequency components;

- Extensive experiments show that MMFNet consistently achieves good performance in a variety of multivariate time series forecasting tasks, with up to a $6.0\%$ reduction in the Mean Squared Error (MSE) compared to the existing models.

## 2 PRELIMINARIES

**Long-term Time Series Forecasting.** LTSF involves predicting future values over an extended time horizon based on previously observed multivariate time series data. The LTSF problem can be formulated as:

$$\hat{x}_{t+1:t+H} = f(x_{t-L+1:t}), \tag{1}$$

where $x_{t-L+1:t} \in \mathbb{R}^{L \times C}$ denotes the historical observation window, and $\hat{x}_{t+1:t+H} \in \mathbb{R}^{H \times C}$ represents the predicted future values. In this formulation, $L$ is the length of the historical window, $H$ is the forecast horizon, and $C$ denotes the number of features or channels. As the forecast horizon $H$ increases, the models face challenges to accurately capture both long-term and short-term dependencies within the time series.

**Single-Scale Frequency Transformation (SFT).** SFT refers to the process of converting the time-domain data into the frequency domain at a single, global scale without segmenting the time series. Such a transformation is typically performed using methods, such as the Fast Fourier Transform

(FFT), which efficiently computes the Discrete Fourier Transform (DFT). SFT decomposes the entire signal into sinusoidal components, enabling the analysis of its frequency content. Each frequency component can be expressed as:

$$X_k = |X_k|e^{j\phi_k}, \tag{2}$$

where $|X_k|$ represents the amplitude and $\phi_k$ the phase of the $k$-th frequency component. While the frequency decomposition provides valuable insights into periodic patterns and trends, traditional approaches assume stationarity and operate on a global scale, limiting their capacity to capture the complex, non-stationary characteristics frequently observed in real-world time series. Current frequency-based LTSF models, such as FITS (Xu et al., 2024), implement this method by performing frequency domain interpolation at a single scale, which can be formulated as:

$$\tilde{x}_{t+1:t+H} = g(\mathcal{F}(x_{t-L+1:t})), \tag{3}$$

where $\mathcal{F}$ denotes the Fourier transform, and $g$ represents the filtering operation applied uniformly across the signal. Although SFT is capable of capturing broad temporal patterns, such as long-term trends through low-pass filtering or short-term fluctuations through high-pass filtering, its global application treats the entire signal uniformly. This uniform treatment may result in the loss of important local temporal variations and non-stationary behaviors occurring at different scales.

## 3 METHOD

### 3.1 OVERVIEW

To overcome the limitations of SFT, we propose the Multi-scale Masked Frequency Transformation (MMFT). MMFT performs frequency decomposition across multiple temporal scales, enabling the model to capture both global and local temporal patterns. Formally, the MMFT problem can be expressed as:

$$\tilde{x}_{t+1:t+H} = h(\{\mathcal{F}_s(x_{t-L+1:t})\}_{s=1}^S), \tag{4}$$

where $\mathcal{F}_s$ denotes the frequency transformation at scale $s$, and $h$ represents the aggregation and filtering operation applied to the learnable frequency masks at various scales. Unlike SFT, which applies a single transformation to the entire time series, MMFT divides the signal into multiple scales, each subjected to frequency decomposition. At each scale, a learnable frequency mask is applied to retain the most informative frequency components while selectively discarding noise. This multi-scale approach allows the model to adapt to non-stationary signals, capturing complex dependencies that span different temporal ranges. By leveraging frequency decomposition at multiple scales and applying adaptive masks, MMFT enhances long-term forecasting accuracy by focusing on both short-term fluctuations and long-term trends within the data. This method increases the model's flexibility and robustness, particularly for non-stationary and multivariate time series. Further analysis of the differences between SFT and MMFT can be found in Appendix A.

MMFNet enhances time series forecasting by incorporating the proposed MMFT method to capture intricate frequency features across different scales. The overall architecture of MMFNet is depicted in Figure 1. The model comprises three key components: Multi-scale Frequency Decomposition, Masked Frequency Interpolation, and Spectral Inversion. Multi-scale Frequency Decomposition normalizes the input time series, divides it into segments of varying scales, and transforms these segments into the frequency domain using the DCT. Masked Frequency Interpolation applies a self-adaptive, learnable mask to filter out irrelevant frequency components, followed by a linear transformation of the filtered frequency domain segments. Finally, Spectral Inversion converts the processed frequency components back into the time domain via the Inverse Discrete Cosine Transform (iDCT) (Ahmed et al., 1974). The outputs from different scales are then aggregated, resulting in a refined signal that preserves the essential characteristics of the original input.

### 3.2 MULTI-SCALE FREQUENCY DECOMPOSITION

The core concept of Multi-scale Frequency Decomposition lies in applying frequency domain transformations to time series sequences at multiple scales. This approach enables the model to capture both global patterns and fine-grained temporal dynamics by analyzing the data across various segment levels. Multi-scale Frequency Decomposition consists of two fundamental steps: fragmentation and decomposition. Details about the overall workflow can be seen in Appendix B.1.

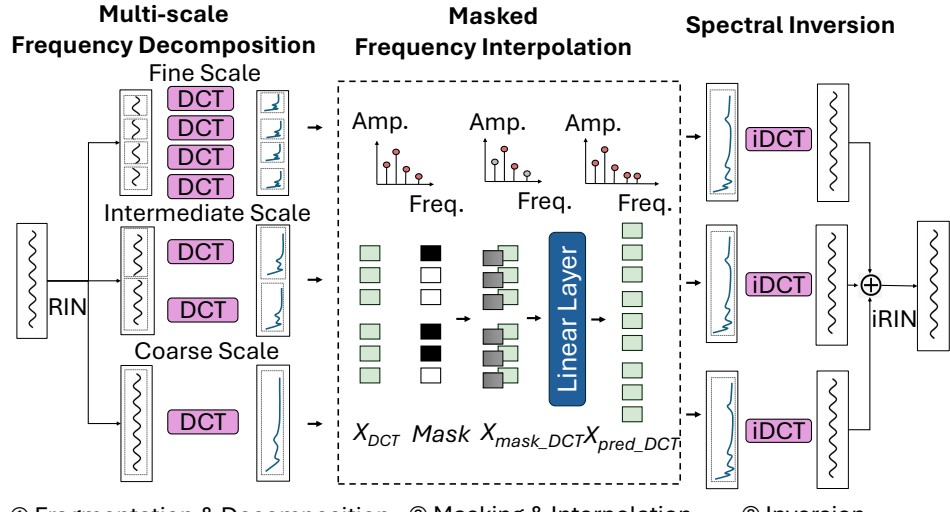

Figure 1: MMFNet Architecture. MMFNet consists of the following key components: ① The input time series is first normalized to have zero mean using Reversible Instance-wise Normalization (RIN) (Lai et al., 2021). The multi-scale frequency decomposition process then divides the time series instance $X$ into fine, intermediate, and coarse-scale segments, which are subsequently transformed into the frequency domain via the Discrete Cosine Transform (DCT). ② A learnable mask is applied to the frequency segments, followed by a linear layer that predicts the transformed frequency components. ③ Finally, the predicted frequency segments from each scale are transformed back into the time domain, merged, and denormalized using inverse RIN (iRIN).

**Fragmentation.** This step decomposes the time series data into segments of varying lengths to capture features across multiple scales. Specifically, the input sequence $X$ is first normalized using RIN (Lai et al., 2021) and then partitioned into three sets of segments: fine-scale, intermediate-scale, and coarse-scale segments. Fine-scale segments ($\boldsymbol{X}^{fine}$) consist of shorter segments that capture detailed, high-frequency components of the time series, enabling the detection of intricate patterns and anomalies that may be missed in longer segments. Intermediate-scale segments ($\boldsymbol{X}^{intermediate}$) are of moderate length and are designed to capture intermediate-level patterns and trends, striking a balance between the fine and coarse segments. Coarse-scale segments ($\boldsymbol{X}^{coarse}$) comprise longer segments that capture broader, low-frequency trends and overarching patterns within the data. This multi-scale fragmentation allows the model to effectively capture and leverage patterns across different temporal scales.

**Decomposition.** This step converts the multi-scale time-domain segments into their corresponding frequency components to capture frequency patterns across various temporal scales. For each segment, the DCT is applied to extract frequency domain representations. Specifically, the fine-scale segments in $\boldsymbol{X}^{fine}$ are transformed into $\boldsymbol{X}_{DCT}^{fine}$, the intermediate-scale segments in $\boldsymbol{X}^{intermediate}$ are converted into $\boldsymbol{X}_{DCT}^{intermediate}$, and the coarse-scale segments in $\boldsymbol{X}^{coarse}$ are transformed into $\boldsymbol{X}_{DCT}^{coarse}$.

The DCT for each segment is computed using the following formula:

$$X_k = \sum_{n=0}^{N-1} x_n \cos\left(\frac{\pi}{N}\left(n + \frac{1}{2}\right)k\right),\qquad(5)$$

where $x_n$ represents the time-domain signal values, $N$ is the segment length, and $k$ denotes the frequency component. The resulting coefficients $X_k$ represent the frequency components of the segment. This transformation enables MMFNet to capture and analyze patterns at multiple temporal scales in the frequency domain, thereby enhancing its ability to recognize and interpret complex patterns in time series data.

### 3.3 MASKED FREQUENCY INTERPOLATION

Masked Frequency Interpolation leverages a learnable mask to adaptively filter frequency components across different scales in the frequency domain, followed by reconstruction through a linear layer neural network. This approach enables the model to learn and apply scale-specific filtering strategies tailored to diverse datasets. The process consists of two primary steps: Masking and Interpolation.

**Masking.** Traditional methods often employ fixed low-pass filters with a predefined cutoff frequency to filter frequency components. These approaches assume that certain frequencies are universally important or irrelevant across the entire time series, an assumption that may not hold for non-stationary data where the relevance of frequency components varies over time. Moreover, over-filtering can lead to the loss of critical details, resulting in oversimplified representations and diminished model performance in tasks such as forecasting and signal analysis. To address these limitations, MMFNet employs an adaptive masking technique to capture dynamic behaviors in the frequency domain. Given the frequency segments $\boldsymbol{X}_{DCT}$, a learnable mask is generated to adaptively filter the frequency components. The mask adjusts the significance of different frequency components by attenuating or emphasizing them based on their relevance to the task. This filtering process is applied via element-wise multiplication, represented as:

$$\boldsymbol{X}_{mask\_DCT} = \boldsymbol{X}_{DCT} \odot M, \tag{6}$$

where $\odot$ denotes element-wise multiplication, $M$ represents the learnable mask, and $\boldsymbol{X}_{mask\_DCT}$ is the resulting masked frequency representation. During training, the mask is iteratively updated based on the loss function, allowing MMFNet to focus on the most relevant aspects of the frequency domain representation. This adaptive mechanism improves the model's capacity to capture meaningful patterns while minimizing the influence of irrelevant or noisy information.

**Interpolation.** In this step, the masked frequency segments $\boldsymbol{X}_{mask\_DCT}$ are transformed into predicted frequency domain segments $\boldsymbol{X}_{pred\_DCT}$ through a linear layer. This linear transformation maps the filtered frequency components to the target frequency representations aligned with the model's forecasting objectives. Specifically, a fully connected (dense) layer is applied to the masked frequency components, and this operation can be expressed as:

$$\boldsymbol{X}_{pred\_DCT} = W \cdot \boldsymbol{X}_{mask\_DCT} + b, \tag{7}$$

where $W$ denotes the weight matrix of the linear layer, and $b$ is the bias term. The linear layer is designed to learn a projection that aligns the filtered frequency components with the target prediction space. This transformation further refines the frequency domain information, producing $\boldsymbol{X}_{pred\_DCT}$, which is essential for reconstructing accurate time-domain predictions. By leveraging the refined frequency information and reducing the influence of irrelevant frequency components, this step improves the overall prediction accuracy.

### 3.4 SPECTRAL INVERSION

The final process, Spectral Inversion, transforms the interpolated frequency components back into the time domain using the iDCT, reversing the earlier DCT process. The iDCT is applied individually to the predicted frequency domain segments $\boldsymbol{X}_{pred\_DCT}^{fine}$, $\boldsymbol{X}_{pred\_DCT}^{intermediate}$, and $\boldsymbol{X}_{pred\_DCT}^{coarse}$. The iDCT for a segment is given by the following formula:

$$x_n = \frac{1}{2}x_0 + \sum_{k=1}^{N-1} X_k \cos\left(\frac{\pi}{N}\left(n + \frac{1}{2}\right)k\right), \tag{8}$$

where $x_n$ represents the time-domain signal values, $X_k$ are the frequency components, and $N$ denotes the segment length. This equation reconstructs the time-domain signal by summing the contributions of each frequency component (Davis & Marsaglia, 1984).

After performing the iDCT separately for each scale, the resulting time-domain signals are combined to merge the multi-scale frequency information. The combination is achieved by averaging the

reconstructed signals from the fine, intermediate, and coarse scales. The final signal $Y$ is computed using the average function as:

$$Y = \text{Average} \left( X_{time}^{fine}, X_{time}^{intermediate}, X_{time}^{coarse} \right), \tag{9}$$

where $X_{time}^{fine}$, $X_{time}^{intermediate}$, and $X_{time}^{coarse}$ are the time-domain signals obtained after applying the iDCT to the respective scales.

This integration step ensures that the multi-scale frequency information is effectively averaged, preserving the key characteristics of the original input while incorporating the enhanced interpolation achieved through the masked frequency filtering.

## 4 EXPERIMENT

In this section, we evaluate MMFNet with several LTSF benchmark datasets across a range of forecast horizons. We also conduct ablation studies to assess the impact of MMFT and our frequency masking techniques. Finally, we evaluate MMFNet's performance in ultra-long-term forecasting scenarios.

### 4.1 EXPERIMENTAL SETUP

**Datasets.** We perform experiments with seven widely-used LTSF datasets: ETTh1, ETTh2, ETTm1, ETTm2, Weather, Electricity, and Traffic. More details on those datasets can be found in Appendix B.2.

**Baselines.** We compare MMFNet against several state-of-the-art models, including FEDformer (Zhou et al., 2022b), TimesNet (Wu et al., 2023), TimeMixer (Wang et al., 2024), and PatchTST (Nie et al., 2024). In addition, we compare MMFNet against several lightweight models, including DLinear (Zeng et al., 2023), FITS (Xu et al., 2024), and SparseTSF (Lin et al., 2024). More details on our baseline models can be found in Appendix B.3.

**Environment.** All experiments are implemented using PyTorch (Paszke et al., 2019) and run on a single NVIDIA GeForce RTX 4090 GPU with 24GB of memory.

### 4.2 PERFORMANCE ON LTSF BENCHMARKS

The experimental results offer several key insights into MMFNet's performance across a range of datasets and forecast horizons. As Table 6 shows, MMFNet demonstrates superior performance on the ETT dataset and consistently achieves the best results even at extended forecasting horizons. Additionally, it maintains strong performance across a range of channel numbers and sampling rates.

**Performance on the ETT Dataset.** As Table 6 shows, MMFNet consistently outperforms other models across all forecast horizons on the ETTh1, ETTh2, and ETTm2 datasets. For example, on ETTh1, compared with other baseline models, MMFNet achieves the best MSE results of $0.359$, $0.396$, $0.409$, and $0.419$ at forecast horizons of $96$, $192$, $336$, and $720$, respectively. Moreover, it demonstrates a $4.2\%$ MSE reduction ($+0.018$) at the forecast horizon of $336$ on ETTh1 and a $5.1\%$ MSE reduction ($+0.018$) at the forecast horizon of $336$ on ETTh2. This consistent performance highlights MMFNet's ability to effectively capture both short-term fluctuations and long-term dependencies in time series data, positioning it as a versatile model for a wide variety of LTSF tasks.

**Performance at the Extended Horizon.** As Table 6 shows, at the extended forecast horizon of $720$, MMFNet consistently achieves the highest predictive accuracy across all datasets, except for Traffic where it ranks second. Notably, MMFNet demonstrates significant improvements over baseline models, achieving MSE reductions of $4.6\%$ ($+0.019$) on ETTm1 and $6.0\%$ ($+0.021$) on ETTm2 at forecast horizon $720$ compared to the second-best models. These results highlight the robustness of MMFNet in addressing long-term forecasting tasks.

Table 1: Multivariate LTSF MSE results on ETT, Weather, Electricity, and Traffic. The best result is emphasized in **bold**, while the second-best is underlined. "Imp." represents the improvement between MMFNet and either the best or second-best result, with a higher "Imp." indicating greater improvement.

| Models | | MMFNet | FITS | SparseTSF | DLinear | PatchTST | TimeMixer | TimesNet | iTransformer | FEDformer | Imp. |
|---|---|---|---|---|---|---|---|---|---|---|---|
| Data | Horizon | (ours) | (2024) | (2024) | (2023) | (2023) | (2024) | (2023) | (2023) | (2022) | |
| ETTh1 | 96 | **0.359** | 0.372 | 0.362 | 0.384 | 0.385 | 0.380 | 0.384 | 0.386 | 0.375 | +0.003 |
| | 192 | **0.396** | 0.404 | 0.403 | 0.443 | 0.413 | 0.413 | 0.436 | 0.441 | 0.427 | +0.006 |
| | 336 | **0.409** | 0.427 | 0.434 | 0.446 | 0.440 | 0.445 | 0.491 | 0.487 | 0.459 | +0.018 |
| | 720 | **0.419** | 0.424 | 0.426 | 0.504 | 0.456 | 0.491 | 0.521 | 0.503 | 0.484 | +0.005 |
| ETTh2 | 96 | **0.263** | 0.271 | 0.294 | 0.282 | 0.274 | 0.281 | 0.340 | 0.297 | 0.340 | +0.008 |
| | 192 | **0.317** | 0.331 | 0.339 | 0.340 | 0.338 | 0.356 | 0.402 | 0.380 | 0.433 | +0.014 |
| | 336 | **0.336** | 0.354 | 0.359 | 0.414 | 0.367 | 0.371 | 0.452 | 0.428 | 0.508 | +0.018 |
| | 720 | **0.376** | 0.377 | 0.383 | 0.588 | 0.391 | 0.403 | 0.462 | 0.427 | 0.480 | +0.001 |
| ETTm1 | 96 | 0.307 | 0.303 | 0.314 | 0.301 | **0.292** | 0.315 | 0.338 | 0.334 | 0.362 | -0.015 |
| | 192 | 0.334 | 0.337 | 0.343 | 0.335 | **0.330** | 0.339 | 0.374 | 0.377 | 0.393 | -0.004 |
| | 336 | **0.358** | 0.366 | 0.369 | 0.371 | 0.365 | 0.366 | 0.410 | 0.426 | 0.442 | +0.007 |
| | 720 | **0.396** | 0.415 | 0.418 | 0.426 | 0.419 | 0.423 | 0.478 | 0.491 | 0.483 | +0.019 |
| ETTm2 | 96 | **0.160** | 0.162 | 0.165 | 0.171 | 0.163 | 0.176 | 0.187 | 0.180 | 0.189 | +0.002 |
| | 192 | **0.212** | 0.216 | 0.218 | 0.237 | 0.219 | 0.226 | 0.249 | 0.250 | 0.256 | +0.004 |
| | 336 | **0.259** | 0.268 | 0.272 | 0.294 | 0.276 | 0.276 | 0.321 | 0.311 | 0.326 | +0.009 |
| | 720 | **0.327** | 0.348 | 0.352 | 0.426 | 0.368 | 0.372 | 0.408 | 0.412 | 0.437 | +0.021 |
| Weather | 96 | 0.153 | **0.143** | 0.172 | 0.174 | 0.151 | 0.159 | 0.172 | 0.174 | 0.246 | -0.010 |
| | 192 | 0.194 | **0.186** | 0.215 | 0.217 | 0.195 | 0.202 | 0.219 | 0.221 | 0.292 | -0.008 |
| | 336 | 0.241 | **0.236** | 0.263 | 0.262 | 0.249 | 0.281 | 0.280 | 0.278 | 0.378 | -0.005 |
| | 720 | **0.302** | 0.307 | 0.318 | 0.332 | 0.321 | 0.335 | 0.365 | 0.358 | 0.447 | +0.005 |
| Electricity | 96 | 0.131 | 0.134 | 0.138 | 0.140 | **0.129** | 0.158 | 0.168 | 0.148 | 0.188 | -0.002 |
| | 192 | **0.146** | 0.149 | 0.151 | 0.153 | 0.149 | 0.174 | 0.184 | 0.162 | 0.197 | +0.003 |
| | 336 | **0.162** | 0.165 | 0.166 | 0.169 | 0.166 | 0.190 | 0.198 | 0.178 | 0.212 | +0.003 |
| | 720 | **0.199** | 0.203 | 0.205 | 0.204 | 0.210 | 0.229 | 0.220 | 0.225 | 0.244 | +0.004 |
| Traffic | 96 | 0.381 | 0.385 | 0.389 | 0.413 | **0.366** | 0.380 | 0.593 | 0.395 | 0.573 | -0.015 |
| | 192 | 0.394 | 0.397 | 0.398 | 0.423 | **0.388** | 0.397 | 0.617 | 0.417 | 0.611 | -0.006 |
| | 336 | 0.408 | 0.410 | 0.411 | 0.437 | **0.398** | 0.418 | 0.629 | 0.433 | 0.621 | -0.010 |
| | 720 | 0.446 | 0.448 | 0.448 | 0.466 | 0.457 | **0.436** | 0.640 | 0.467 | 0.630 | -0.010 |

**Performance in Low-Channel, Low-Sampling Rate Scenarios.** As Table 6 shows, in scenarios involving datasets with fewer channels (7 channels) and lower sampling rates (1-hour intervals), such as in the ETTh1 and ETTh2 datasets, linear models like FITS, SparseTSF, and DLinear exhibit strong performance. For example, on ETTh2, FITS achieves the MSE results of 0.271, 0.331, 0.354, and 0.377 at forecast horizons of 96, 192, 336, and 720, respectively. MMFNet continues to surpass these models on ETTh2 by achieving the MSE results of 0.263, 0.317, 0.336, and 0.376 at forecast horizons of 96, 192, 336, and 720, respectively. This suggests that multi-scale frequency decomposition methods are particularly well-suited for datasets with fewer channels and broader time intervals between measurements.

**Performance in High-Channel Scenarios.** As Table 6 shows, for datasets with larger numbers of channels, such as Electricity (321 channels, 1-hour sampling rate) and Traffic (862 channels, 1-hour sampling rate), MMFNet and FITS consistently demonstrate strong performance. Despite the increased complexity that arises from higher channel counts. For example, on Electricity, MMFnet achieves the best MSE results of 0.146, 0.162, and 0.199 at forecast horizons of 192, 336, and 720, respectively. MMFNet's multi-scale frequency decomposition enables it to effectively model complex temporal dependencies while maintaining high predictive accuracy. While PatchTST performs better on the traffic dataset, it leverages a patching transformer mechanism rather than a purely linear frequency-based approach, distinguishing it from MMFNet and FITS in terms of the model architecture. This further indicates that more sophisticated decomposition methods are required for lightweight models to handle high-channel scenarios effectively.

**Performance in High-Sampling Rate Scenarios.** As Table 6 shows, for datasets with higher sampling rates, such as Weather (21 channels, 10-minute sampling rate), ETTm1 and ETTm2 (7 channels, 15-minute sampling rate), MMFNet and FITS consistently demonstrate strong performance. For example, on ETTm2, MMFnet achieves the best MSE results of 0.160, 0.212, 0.259, and 0.327 at forecast horizons of 96, 192, 336, and 720, respectively. Despite the increased complexity that arises from a faster sampling rate, MMFNet's multi-scale frequency decomposition enables it to effectively model complex temporal dependencies while maintaining high predictive accuracy.

### 4.3 COMPARISONS BETWEEN MMFT AND SFT

Table 2: MSE values of MMFNet when it uses SFT and MMFT on the ETT dataset. SFT denotes the standard single-scale frequency decomposition approach. MFT refers to the masked frequency transformation with fragmentation applied at a single scale, where $N_{seg}$ specifies the segment length. MMFT denotes the full MMFT method, which performs frequency decomposition with multi-scale fragmentation. "Imp." indicates the improvement of MMFT over SFT.

| Dataset | ETTh1 | | | | ETTh2 | | | |
|---|---|---|---|---|---|---|---|---|
| Horizon | 96 | 192 | 336 | 720 | 96 | 192 | 336 | 720 |
| SFT | 0.372 | 0.404 | 0.427 | 0.424 | 0.271 | 0.331 | 0.354 | 0.377 |
| MFT ($N_{seg} = 24$) | 0.362 | 0.400 | 0.412 | 0.421 | 0.264 | 0.317 | 0.336 | 0.376 |
| MFT ($N_{seg} = 120$) | 0.366 | 0.401 | 0.426 | 0.423 | 0.265 | 0.317 | 0.336 | 0.376 |
| MFT ($N_{seg} = 360$) | 0.366 | 0.403 | 0.418 | 0.425 | 0.265 | 0.317 | 0.340 | 0.376 |
| MMFT | **0.359** | **0.396** | **0.409** | **0.419** | **0.263** | **0.317** | **0.336** | **0.376** |
| Imp.(MMFT over SFT) | +0.013 | +0.008 | +0.018 | +0.005 | +0.008 | +0.014 | +0.018 | +0.001 |

To evaluate the effectiveness of the MMFT method (see Section 3.1), we perform experiments using the ETT dataset. Both SFT and MMFT incorporate the same adaptive masking strategy to ensure fair and consistent comparisons. SFT applies FFT to the entire time series without fragmentation, while MFT introduces a single-scale fragmentation, and MMFT performs a multi-scale fragmentation. The results presented in Table 2 reveal two important insights.

First, fragmentation consistently enhances frequency domain decomposition. On the ETTh1 dataset, MFT ($N_{seg} = 360$) achives the MSE results of 0.160, 0.212, 0.259, and 0.327 at forecast horizons of 96, 192, 336, and 720, respectively. MFT delivers the most significant gains observed at a segment length of 24 with a $4.2\%$ MSE reduction (+0.018) at then forecast horizon of 336. This improvement suggests that segmenting the time series into smaller segments enables MFT to capture localized frequency features more effectively.

Second, MMFT, leveraging multi-scale decomposition, consistently delivers superior results compared to both SFT and single-scale MFT. On the ETTh2 dataset, MMFT achives the MSE results of 0.263, 0.317, 0.336, and 0.376 at forecast horizons of 96, 192, 336, and 720, respectively. At the forecast horizon of 336, MMFT achieves substantial reductions in MSE, including a 0.018 improvement over SFT. These results suggest that the multi-scale decomposition employed by MMFT allows for the capture of a broader range of frequency patterns, leading to more accurate predictions, particularly in long-term forecasting scenarios.

### 4.4 EFFECTIVENESS OF MASKING

Table 3: MSE results for multivariate LTSF with MMFNet on the ETT dataset with or without the masking module. "Mask" refers to results with the masking module, while "w/o Mask" refers to results without it. "Imp." denotes the improvement enabled by the masking module.

| Dataset | ETTh1 | | | | ETTh2 | | | | Electricity | | | | Traffic | | | |
|---|---|---|---|---|---|---|---|---|---|---|---|---|---|---|---|---|
| Horizon | 96 | 192 | 336 | 720 | 96 | 192 | 336 | 720 | 96 | 192 | 336 | 720 | 96 | 192 | 336 | 720 |
| w/o Mask | 0.372 | 0.405 | 0.410 | 0.420 | 0.269 | 0.319 | 0.339 | 0.376 | 0.312 | 0.338 | 0.360 | 0.397 | 0.166 | 0.218 | 0.264 | 0.330 |
| Mask | **0.359** | **0.396** | **0.409** | **0.419** | **0.263** | **0.317** | **0.336** | **0.376** | **0.307** | **0.334** | **0.358** | **0.396** | **0.160** | **0.212** | **0.259** | **0.327** |
| Imp. | +0.013 | +0.009 | +0.001 | +0.001 | +0.006 | +0.002 | +0.003 | +0.000 | +0.005 | +0.003 | +0.002 | +0.001 | +0.006 | +0.006 | +0.005 | +0.003 |

To evaluate the effectiveness of the self-adaptive masking mechanism, we compare MMFNet's performance on the ETT dataset with and without the masking module across four forecast horizons: 96, 192, 336, and 720. As Table 3 lists, MMFNet with masking consistently outperforms the version without masking across all horizons. The most notable improvements occur at the horizon 96 with a $3.5\%$ MSE reduction on ETTh1 (+0.013) and a $2.2\%$ MSE reduction on ETTh2 (+0.006). With the Electricity dataset, the largest improvement is at horizon 96 with an improvement of +0.005. Similarly, the largest improvement is at horizon 192 with an improvement of +0.006 on the Traffic dataset. The results show that the self-adaptive masking mechanism which filters out frequency noise at different scales consistently enhances forecasting accuracy across various datasets and forecast horizons.

## 4.5 Performance on Ultra-long-term Time Series Forecasting

Table 4: MSE results for multivariate ultra long-term time series forecasting with MMFNet. The best result is emphasized in **bold**, while the second-best is underlined. "Imp." represents the improvement between MMFNet and either the best or second-best result, with a higher "Imp." value indicating greater improvement.

| Dataset | ETTm1 | | | | ETTm2 | | | | Electricity | | | | Weather | | | |
|---|---|---|---|---|---|---|---|---|---|---|---|---|---|---|---|---|
| Horizon | 960 | 1200 | 1440 | 1680 | 960 | 1200 | 1440 | 1680 | 960 | 1200 | 1440 | 1680 | 960 | 1200 | 1440 | 1680 |
| DLinear | 0.429 | 0.440 | 0.463 | 0.481 | 0.412 | 0.398 | 0.430 | 0.478 | 0.238 | 0.267 | 0.277 | 0.296 | 0.330 | 0.341 | 0.345 | 0.356 |
| FITS | 0.413 | 0.422 | 0.425 | 0.427 | 0.347 | 0.358 | **0.355** | 0.350 | 0.238 | 0.268 | 0.293 | 0.311 | 0.333 | 0.343 | 0.353 | 0.360 |
| SparseTSF | 0.415 | 0.422 | 0.424 | 0.425 | 0.353 | 0.367 | 0.357 | 0.353 | 0.228 | 0.256 | 0.281 | 0.298 | 0.329 | 0.339 | 0.347 | 0.353 |
| MMFNet(ours) | **0.411** | **0.419** | **0.423** | **0.424** | **0.346** | **0.357** | 0.356 | **0.349** | **0.224** | **0.255** | **0.280** | **0.292** | **0.318** | **0.331** | **0.340** | **0.349** |
| Imp. | +0.002 | +0.003 | +0.001 | +0.001 | +0.001 | +0.001 | -0.001 | +0.001 | +0.004 | +0.001 | +0.001 | +0.004 | +0.011 | +0.008 | +0.005 | +0.004 |

We evaluate MMFNet's performance in ultra-long-term time series forecasting scenarios. Table 4 presents the MSE results for various models applied to multivariate ultra-long-term time series forecasting across four datasets at forecast horizons of 960, 1200, 1440, and 1680. Due to the significant memory requirements of models such as FEDformer, TimesNet, TimeMixer, and PatchTST when forecast horizons are extended, these models exceed GPU memory limitations. Consequently, in this context, we limit the comparison to more lightweight models: DLinear, FITS, SparseTSF, and the proposed MMFNet.

The results show that MMFNet consistently outperforms the existing models across most datasets and forecast horizons. For example, with the ETTh1 dataset, MMFNet achieves the MSE values of 0.411, 0.419, 0.423, and 0.424 at horizons of 960, 1200, 1440, and 1680, respectively. With the Electricity dataset, MMFNet delivers very good performance, particularly at longer horizons, with the MSE values of 0.255 at 1200 and 0.292 at 1680.On the Weather dataset, MMFNet demonstrates superior performance, achieving MSE values of 0.318 at the 960 horizon and 0.331 at the 1200 horizon, representing a 3.3% (+0.011) and 2.4% (+0.008) reduction in MSE compared to the second-best baseline. The results demonstrate the robustness of MMFNet in forecasting multivariate ultra-long-term time series data across various datasets and extended forecast horizons by effectively capturing frequency variations at different scales.

## 5 Related Work

### 5.1 Long-term Time Series Forecasting

LTSF is a critical area in data science and machine learning and focuses on predicting future values over extended periods. Such a task is challenging due to the inherent seasonality, trends, and noise in time series data. In addition, time series data is often complex and high-dimensional Zheng et al. (2024; 2023). Traditional statistical methods, such as ARIMA (Contreras et al., 2003) and Holt-Winters (Chatfield & Yar, 1988), are effective for short-term forecasting but frequently fall short for longer horizons. Machine learning models, such as SVM (Wang & Hu, 2005), Random Forests Breiman (2001), and Gradient Boosting Machines (Natekin & Knoll, 2013), offer improved performance by capturing non-linear relationships but typically require extensive feature engineering. Recently, deep learning models, such as RNNs, LSTMs, GRUs, and Transformer-based models (Informer and Autoformer), have demonstrated notable efficiency in modeling long-term dependencies. Furthermore, the hybrid models that combine statistical methods with machine learning or deep learning techniques have shown improved accuracy. State-of-the-art models, such as FEDformer (Zhou et al., 2022b), FiLM (Zhou et al., 2022a), PatchTST Nie et al. (2024), and SparseTSF, leverage frequency domain transformations and efficient self-attention to improve prediction performance.

### 5.2 Multiscaling Model

In the field of computer vision, several multi-scale Vision Transformers (ViTs) have leveraged hierarchical architectures to generate progressively down-sampled pyramid features. For instance, Multi-Scale Vision Transformers (Fan et al., 2021) enhance the standard Vision Transformer architecture by incorporating multi-scale processing, allowing for improved detail capture across varying

spatial resolutions. Pyramid Vision Transformer (Wang et al., 2021) integrates a pyramid structure within ViTs to facilitate multi-scale feature extraction, while Twins (Dai et al., 2021) combines local and global attention to effectively model multi-scale representations. SegFormer (Xie et al., 2021) introduces an efficient hierarchical encoder that captures both coarse and fine features, and CSWin (Dong et al., 2022) further improves performance by using multi-scale cross-shaped local attention mechanisms. In the context of time series forecasting, TimeMixer (Wang et al., 2024) represents a significant advancement with its fully MLP-based architecture, which employs Past-Decomposable-Mixing and Future-Multipredictor-Mixing blocks. This architecture enables TimeMixer to effectively leverage disentangled multi-scale time series data during both past extraction and future prediction phases.

### 5.3 Time Series Forecasting in the Frequency Domain

Recent advancements in time series analysis have increasingly utilized frequency domain information to reveal underlying patterns. For instance, FNet (Lee-Thorp et al., 2021) adopts an attention-based approach to capture temporal dependencies within the frequency domain, thereby eliminating the need for convolutional or recurrent layers. Models such as FEDformer (Zhou et al., 2022b) and FiLM (Zhou et al., 2022a) improve predictive performance by incorporating frequency domain information as auxiliary features. FITS (Xu et al., 2024) also demonstrates strong predictive capabilities by converting time-domain forecasting tasks into the frequency domain and utilizing low-pass filters to reduce the number of parameters required. However, many of these techniques rely on manual feature engineering to identify dominant periods, which can constrain the amount of information captured and introduce inefficiencies or risks of overfitting.

### 5.4 Masked Modeling

Masked language modeling and its autoregressive variants have emerged as dominant self-supervised learning approaches in natural language processing. These techniques enable large-scale language models to excel in both language understanding and generation by predicting masked or hidden tokens within sentences (Devlin et al., 2018; Radford et al., 2018). In computer vision, early approaches, such as the context encoder (Pathak et al., 2016), involve masking specific regions of an image and predicting the missing pixels, while Contrastive Predictive Coding (van den Oord et al., 2018) uses contrastive learning to improve feature representations. Recent innovations in MIM include models like iGPT (Chen et al., 2020), ViT (Dosovitskiy et al., 2021), and BEiT (Bao et al., 2022), which leverage Vision Transformers and techniques, such as pixel clustering, mean color prediction, and block-wise masking. In the realm of multivariate time series forecasting, masked encoders have recently been employed with notable success in classification and regression tasks (Zerveas et al., 2021). For example, PatchTST uses a masked self-supervised representation learning method to reconstruct the masked patches and showcases its effectiveness in time series data (Nie et al., 2024). However, the application of masked modeling techniques in linear time series forecasting remains relatively under-explored.

## 6 Conclusion

MMFNet significantly advances long-term multivariate forecasting by employing the MMFT approach. Through comprehensive evaluations on benchmark datasets, we have demonstrated that MMFNet consistently outperforms state-of-the-art models in forecasting accuracy, highlighting its robustness in capturing complex data patterns. By effectively integrating multi-scale decomposition with a learnable masked filter, MMFNet captures intricate temporal details while adaptively mitigating noise, making it a versatile and reliable solution for a wide range of LTSF tasks.

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

# A ADVANTAGE OF MMFT OVER SFT

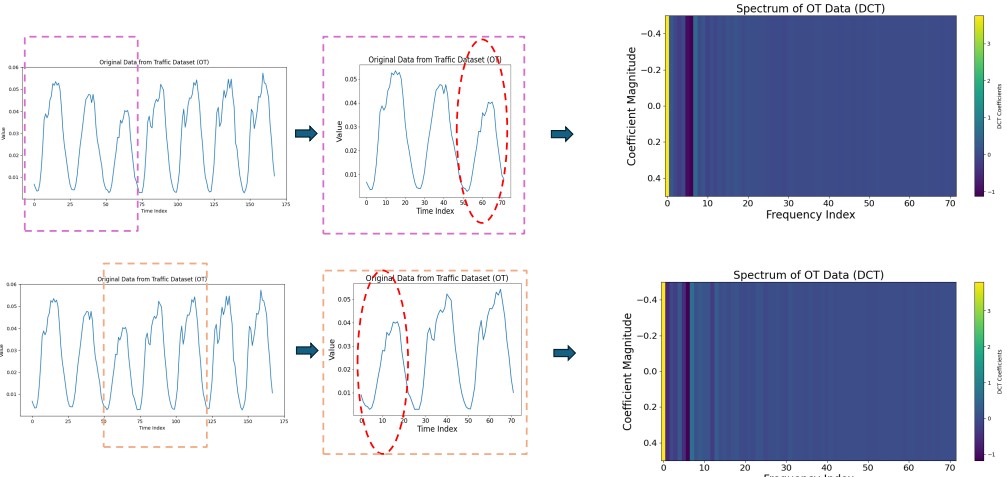

Figure 2: SFT (Different segments produce similar spectra in the frequency domain due to the loss of location information, as highlighted in the red circle). Data is taken from a segment of the Traffic dataset's OT column.

Single-scale frequency domain decomposition provides a global representation of time series data by analyzing the overall frequency spectrum of the entire sequence. While effective for capturing broad trends or global patterns, this method suffers from a significant limitation: the inability to localize specific frequency components to particular segments of the sequence. This drawback is especially problematic for non-stationary time series data, where frequency characteristics evolve over time. For instance, high-frequency noise or transient events might be confined to specific cycles or segments of the sequence. As illustrated in Figure 2, single-scale analysis often produces similar spectra for different segments, losing crucial location-specific information (highlighted by the red circle). This lack of localized detail hinders accurate forecasting, particularly in complex multivariate scenarios where capturing subtle temporal and spectral variations is essential.

To address these challenges, **MMFNet** introduces **Multi-scale Frequency Masking**, which overcomes the limitations of single-scale methods by enabling localized frequency domain analysis. By segmenting the sequence and performing frequency decomposition at multiple scales, MMFNet captures both global patterns and localized high-frequency variations. This approach ensures that critical frequency features—such as transient events or high-frequency noise within specific cycles—are preserved and effectively utilized for forecasting, rather than being obscured in a single-scale global analysis. As shown in Figure 3, MFT enables MMFNet to retain essential location-specific frequency details across segments, allowing for the identification and modeling of hierarchical and nested frequency structures in the data.

MMFT, the core of MMFNet, demonstrates superiority over traditional single-scale frequency decomposition (SFT) by excelling at capturing temporal patterns at fine, intermediate, and coarse-grained scales. Unlike SFT-based models such as FiTS, which often lose location-specific information during global frequency analysis, MMFT preserves this information through its multi-scale approach. By maintaining a hierarchical representation, MMFT ensures that both global trends and localized variations are accurately captured, enhancing the model's ability to manage complex temporal dependencies. Additionally, MMFT incorporates a dynamic masking mechanism, which adaptively filters out irrelevant or noisy frequency components. This ensures that the model focuses on meaningful features while suppressing noise, enhancing its robustness in complex or noisy datasets. In contrast, traditional SFT models lack this adaptive capability, making them less effective in distinguishing signal from noise, especially in datasets with high variability or non-stationarity.

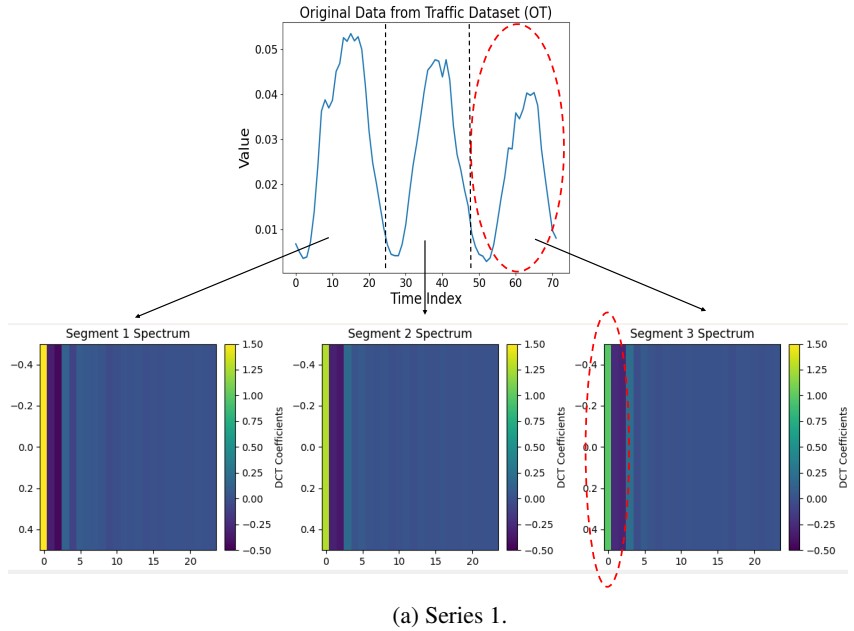

(a) Series 1.

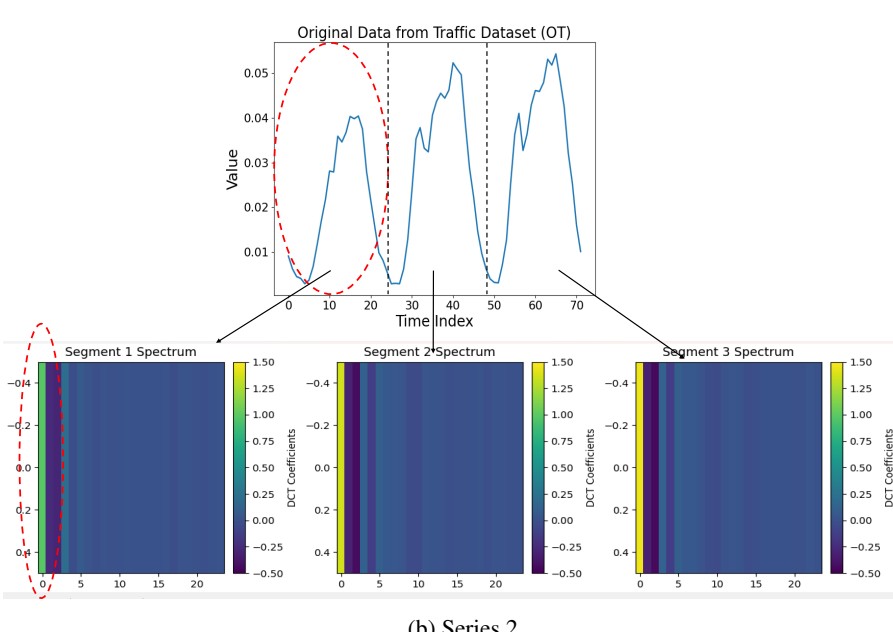

(b) Series 2

Figure 3: MFT (location information is captured).

By combining the advantages of multi-scale pattern representation with adaptive noise filtering, MMFT provides a robust framework for analyzing and forecasting multivariate time series data. Its ability to preserve and exploit multi-scale frequency characteristics ensures superior performance compared to single-scale methods like SFT. MMFNet's innovative approach not only bridges the gap between global and local frequency analysis but also offers a powerful solution for handling complex, dynamic patterns in real-world time series applications.

# B MORE ON MMFNET

## B.1 OVERALL WORKFLOW

The overall workflow of MMFNet is presented in Algorithm 1. The algorithm takes a univariate historical look-back window as input, $x_{t-L+1:t}$, and produces the corresponding forecast, $\hat{x}_{t+1:t+H}$. By incorporating the channel-independent strategy, in which multiple channels are modeled using a shared set of parameters, MMFNet can efficiently extend to multivariate time series forecasting tasks. Such an approach enables the model to leverage its multi-scale frequency decomposition and adaptive masking framework across various input channels to enhance its predictive capabilities in complex multivariate settings.

---

**Algorithm 1** Overall Pseudocode of MMFNet

---

**Require:** Historical look-back window $x_{t-L+1:t} \in \mathbb{R}^L$
**Ensure:** Forecasted output $\hat{x}_{t+1:t+H} \in \mathbb{R}^H$
1: $x_d \leftarrow \text{RIN}(x_{t-L+1:t})$        ▷ Apply Reversible Instance-wise Normalization (RIN)
2: $X_{\text{fine}} \leftarrow \text{Reshape}(x_d, (n_{fine}, s_{fine}))$        ▷ Reshape $x_d$ into a $n_{fine} \times s_{fine}$ matrix
3: $X_{DCT}^{\text{fine}} \leftarrow \text{DCT}(X_{\text{fine}})$        ▷ Apply DCT to each segment with Equation 5
4: $X_{\text{mask\_DCT}}^{\text{fine}} \leftarrow X_{\text{DCT}}^{\text{fine}} \odot \text{Mask}_{fine}$        ▷ Apply the learnable mask
5: $x_{\text{mask\_DCT}}^{\text{fine}} \leftarrow \text{Reshape}(X_{\text{mask\_DCT}}^{\text{fine}})$        ▷ Reshape the matrix back to a sequence of length $L$
6: $x_{\text{pred\_DCT}}^{\text{fine}} \leftarrow \text{Linear}(x_{\text{mask\_DCT}}^{\text{fine}})$        ▷ Apply a linear transformation
7: $x_{\text{fine\_pred}} \leftarrow \text{iDCT}(x_{\text{pred\_DCT}}^{\text{fine}})$        ▷ Apply iDCT to recover the time domain with Equation 8
8: $X_{\text{inter}} \leftarrow \text{Reshape}(x_d, (n_{inter}, s_{inter}))$        ▷ Reshape $x_d$ into a $n_{inter} \times s_{inter}$ matrix
9: $X_{DCT}^{\text{inter}} \leftarrow \text{DCT}(X_{\text{inter}})$        ▷ Apply DCT to each intermediate-scale segment with Equation 5
10: $X_{\text{mask\_DCT}}^{\text{inter}} \leftarrow X_{\text{DCT}}^{\text{inter}} \odot \text{Mask}_{inter}$        ▷ Apply the learnable mask
11: $x_{\text{mask\_DCT}}^{\text{inter}} \leftarrow \text{Reshape}(X_{\text{mask\_DCT}}^{\text{inter}})$        ▷ Reshape the matrix back to a sequence of length $L$
12: $x_{\text{pred\_DCT}}^{\text{inter}} \leftarrow \text{Linear}(x_{\text{mask\_DCT}}^{\text{inter}})$        ▷ Apply a linear transformation
13: $x_{\text{inter\_pred}} \leftarrow \text{iDCT}(x_{\text{pred\_DCT}}^{\text{inter}})$        ▷ Apply iDCT to recover the time domain with Equation 8
14: $X_{\text{coarse}} \leftarrow \text{Reshape}(x_d, (n_{coarse}, s_{coarse}))$        ▷ Reshape $x_d$ into a $n_{coarse} \times s_{coarse}$ matrix
15: $X_{DCT}^{\text{coarse}} \leftarrow \text{DCT}(X_{\text{coarse}})$        ▷ Apply DCT to each coarse-scale segment with Equation 5
16: $X_{\text{mask\_DCT}}^{\text{coarse}} \leftarrow X_{\text{DCT}}^{\text{coarse}} \odot \text{Mask}_{coarse}$        ▷ Apply the learnable mask
17: $x_{\text{mask\_DCT}}^{\text{coarse}} \leftarrow \text{Reshape}(X_{\text{mask\_DCT}}^{\text{coarse}})$        ▷ Reshape the matrix back to a sequence of length $L$
18: $x_{\text{pred\_DCT}}^{\text{coarse}} \leftarrow \text{Linear}(x_{\text{mask\_DCT}}^{\text{coarse}})$        ▷ Apply a linear transformation
19: $x_{\text{coarse\_pred}} \leftarrow \text{iDCT}(x_{\text{pred\_DCT}}^{\text{coarse}})$        ▷ Apply iDCT to recover the time domain with Equation 8
20: $x_M \leftarrow x_{\text{fine\_pred}} + x_{\text{inter\_pred}} + x_{\text{coarse\_pred}} + e_t$        ▷ Combine predictions from all scales and add back the mean
21: $\hat{x}_{t+1:t+H} \leftarrow \text{iRIN}(x_M)$        ▷ Apply inverse Reversible Instance-wise Normalization (iRIN)

---

## B.2 DETAILED DATASET DESCRIPTION

Table 5: Statistics of the datasets.

| Dataset | Traffic | Electricity | Weather | ETTh1 | ETTh2 | ETTm1 | ETTm2 |
|---------|---------|-------------|---------|-------|-------|-------|-------|
| Channels | 862 | 321 | 21 | 7 | 7 | 7 | 7 |
| Sampling Rate | 1 hour | 1 hour | 10 min | 1 hour | 1 hour | 15 min | 15 min |
| Total Timesteps | 17,544 | 26,304 | 52,696 | 17,420 | 17,420 | 69,680 | 69,680 |

Here is a brief description of the datasets used in our experiments.

- The ETT dataset[1] comprises data originally collected for Informer (Zhou et al., 2021), including load and oil temperature measurements recorded at 15-minute intervals between July 2016 and July 2018. The ETTh1 and ETTh2 subsets are sampled at 1-hour intervals, while ETTm1 and ETTm2 are sampled at 15-minute intervals.

---

[1]https://github.com/zhouhaoyi/ETDataset

- The Electricity dataset[2] contains hourly electricity consumption data for 321 customers from 2012 to 2014.

- The Traffic dataset[3] consists of hourly road occupancy rates, collected by various sensors deployed on freeways in the San Francisco Bay area, sourced from the California Department of Transportation.

- The Weather dataset[4] includes local climatological data from nearly $1,600$ locations across the United States, covering a period of four years (2010 to 2013), with data points recorded at 1-hour intervals.

- The Solar-Energy[5] dataset records the solar power production from 137 PV plants in Alabama State, which are sampled every 10 minutes in 2016.

- The Exchange-Rate[6] dataset collects the daily exchange rates of 8 foreign countries from 1990 to 2016.

## B.3 BASELINE MODELS

Here is a brief description of the baseline models used in this paper.

- FEDformer (Zhou et al., 2022b) is a Transformer-based model proposing seasonal-trend decomposition and exploiting the sparsity of time series in the frequency domain. The source code is available at `https://github.com/DAMO-DI-ML/ICML2022-FEDformer`.

- TimesNet (Wu et al., 2023) is a CNN-based model with TimesBlock as a task-general backbone. It transforms 1D time series into 2D tensors to capture intraperiod and interperiod variations. The source code is available at `https://github.com/thuml/TimesNet`.

- TimeMixer (Wang et al., 2024) is a fully MLP-based architecture with PDM and FMM blocks to take full advantage of disentangled multiscale series in both past extraction and future prediction phases. The source code is available at `https://github.com/kwuking/TimeMixer`.

- iTransformer (Wu et al., 2023) is a Transformer based architecture that applies the attention and feed-forward network on the inverted dimensions. The source code is available at `https://github.com/thuml/iTransformer`.

- PatchTST (Nie et al., 2024) is a transformer-based model utilizing patching and CI technique. It also enables effective pre-training and transfer learning across datasets. The source code is available at `https://github.com/yuqinie98/PatchTST`.

- DLinear (Zeng et al., 2023) is an MLP-based model with just one linear layer, which outperforms Transformer-based models in LTSF tasks. The source code is available at `https://github.com/cure-lab/LTSF-Linear`.

- FITS (Xu et al., 2024) is a linear model that manipulates time series data through interpolation in the complex frequency domain. The source code is available at `https://github.com/VEWOXIC/FITS`.

- SparseTSF (Lin et al., 2024) a novel, extremely lightweight model for LTSF, designed to address the challenges of modeling complex temporal dependencies over extended horizons with minimal computational resources. The source code is available at `https://github.com/lss-1138/SparseTSF`.

Table 6: Multivariate LTSF MSE results on ETT, Weather, Electricity, and Traffic. The best result is emphasized in **bold**, while the second-best is underlined. "Imp." represents the improvement between MMFNet and either the best or second-best result, with a higher "Imp." indicating greater improvement.

| Models Data | Horizon | MMFNet (ours) | FITS (2024) | SparseTSF (2024) | DLinear (2023) | PatchTST (2023) | TimeMixer (2024) | TimesNet (2023) | iTransformer (2023) | FEDformer (2022) | Imp. |
|---|---|---|---|---|---|---|---|---|---|---|---|
| ETTh1 | 96 | **0.359** | 0.372 | 0.362 | 0.384 | 0.385 | 0.380 | 0.384 | 0.386 | 0.375 | +0.003 |
| | 192 | **0.396** | 0.404 | 0.403 | 0.443 | 0.413 | 0.413 | 0.436 | 0.441 | 0.427 | +0.006 |
| | 336 | **0.409** | 0.427 | 0.434 | 0.446 | 0.440 | 0.445 | 0.491 | 0.487 | 0.459 | +0.018 |
| | 720 | **0.419** | 0.424 | 0.426 | 0.504 | 0.456 | 0.491 | 0.521 | 0.503 | 0.484 | +0.005 |
| ETTh2 | 96 | **0.263** | 0.271 | 0.294 | 0.282 | 0.274 | 0.281 | 0.340 | 0.297 | 0.340 | +0.008 |
| | 192 | **0.317** | 0.331 | 0.339 | 0.340 | 0.338 | 0.356 | 0.402 | 0.380 | 0.433 | +0.014 |
| | 336 | **0.336** | 0.354 | 0.359 | 0.414 | 0.367 | 0.371 | 0.452 | 0.428 | 0.508 | +0.018 |
| | 720 | **0.376** | 0.377 | 0.383 | 0.588 | 0.391 | 0.403 | 0.462 | 0.427 | 0.480 | +0.001 |
| ETTm1 | 96 | 0.307 | 0.303 | 0.314 | 0.301 | **0.292** | 0.315 | 0.338 | 0.334 | 0.362 | -0.015 |
| | 192 | 0.334 | 0.337 | 0.343 | 0.335 | **0.330** | 0.339 | 0.374 | 0.377 | 0.393 | -0.004 |
| | 336 | **0.358** | 0.366 | 0.369 | 0.371 | 0.365 | 0.366 | 0.410 | 0.426 | 0.442 | +0.007 |
| | 720 | **0.396** | 0.415 | 0.418 | 0.426 | 0.419 | 0.423 | 0.478 | 0.491 | 0.483 | +0.019 |
| ETTm2 | 96 | **0.160** | 0.162 | 0.165 | 0.171 | 0.163 | 0.176 | 0.187 | 0.180 | 0.189 | +0.002 |
| | 192 | **0.212** | 0.216 | 0.218 | 0.237 | 0.219 | 0.226 | 0.249 | 0.250 | 0.256 | +0.004 |
| | 336 | **0.259** | 0.268 | 0.272 | 0.294 | 0.276 | 0.276 | 0.321 | 0.311 | 0.326 | +0.009 |
| | 720 | **0.327** | 0.348 | 0.352 | 0.426 | 0.368 | 0.372 | 0.408 | 0.412 | 0.437 | +0.021 |
| Weather | 96 | 0.153 | **0.143** | 0.172 | 0.174 | 0.151 | 0.159 | 0.172 | 0.174 | 0.246 | -0.010 |
| | 192 | 0.194 | **0.186** | 0.215 | 0.217 | 0.195 | 0.202 | 0.219 | 0.221 | 0.292 | -0.008 |
| | 336 | 0.241 | **0.236** | 0.263 | 0.262 | 0.249 | 0.281 | 0.280 | 0.278 | 0.378 | -0.005 |
| | 720 | **0.302** | 0.307 | 0.318 | 0.332 | 0.321 | 0.335 | 0.365 | 0.358 | 0.447 | +0.005 |
| Electricity | 96 | 0.131 | 0.134 | 0.138 | 0.140 | **0.129** | 0.158 | 0.168 | 0.148 | 0.188 | -0.002 |
| | 192 | **0.146** | 0.149 | 0.151 | 0.153 | 0.149 | 0.174 | 0.184 | 0.162 | 0.197 | +0.003 |
| | 336 | **0.162** | 0.165 | 0.166 | 0.169 | 0.166 | 0.190 | 0.198 | 0.178 | 0.212 | +0.003 |
| | 720 | **0.199** | 0.203 | 0.205 | 0.204 | 0.210 | 0.229 | 0.220 | 0.225 | 0.244 | +0.004 |
| Traffic | 96 | 0.381 | 0.385 | 0.389 | 0.413 | **0.366** | 0.380 | 0.593 | 0.395 | 0.573 | -0.015 |
| | 192 | 0.394 | 0.397 | 0.398 | 0.423 | **0.388** | 0.397 | 0.617 | 0.417 | 0.611 | -0.006 |
| | 336 | 0.408 | 0.410 | 0.411 | 0.437 | **0.398** | 0.418 | 0.629 | 0.433 | 0.621 | -0.010 |
| | 720 | 0.446 | 0.448 | 0.448 | 0.466 | 0.457 | **0.436** | 0.640 | 0.467 | 0.630 | -0.010 |
| Health | 24 | 1.931 | 2.149 | 1.981 | 2.088 | **1.916** | 2.545 | 2.317 | 2.008 | 2.624 | -0.015 |
| | 36 | 1.953 | 2.681 | 1.980 | 1.963 | **1.834** | 2.367 | 1.972 | 2.239 | 2.516 | -0.119 |
| | 48 | 2.058 | 2.912 | **1.954** | 2.130 | 2.107 | 3.072 | 2.238 | 2.187 | 2.505 | -0.104 |
| | 60 | **1.937** | 2.179 | 1.981 | 2.368 | 2.023 | 2.988 | 2.027 | 2.084 | 2.742 | +0.044 |
| Solar | 24 | 0.191 | 0.195 | 0.211 | 0.290 | 0.265 | 0.189 | 0.273 | 0.203 | 0.286 | -0.002 |
| | 36 | 0.212 | 0.216 | 0.225 | 0.320 | 0.288 | 0.222 | 0.297 | 0.233 | 0.291 | -0.006 |
| | 48 | **0.230** | 0.232 | 0.241 | 0.353 | 0.301 | 0.231 | 0.320 | 0.248 | 0.354 | +0.001 |
| | 60 | 0.236 | 0.242 | 0.241 | 0.357 | 0.295 | **0.223** | 0.320 | 0.249 | 0.380 | -0.013 |
| Exchange | 24 | **0.083** | 0.086 | 0.105 | 0.087 | 0.087 | 0.090 | 0.107 | 0.086 | 0.248 | +0.003 |
| | 36 | **0.175** | 0.180 | 0.193 | 0.251 | 0.183 | 0.187 | 0.226 | 0.177 | 0.271 | +0.002 |
| | 48 | **0.329** | 0.333 | 0.358 | 0.403 | 0.390 | 0.353 | 0.367 | 0.331 | 0.460 | -0.002 |
| | 60 | **0.928** | 0.941 | 0.954 | 1.364 | 1.038 | 0.934 | 0.964 | 0.970 | 1.195 | +0.006 |

# C    MORE EXPERIMENTAL RESULTS

## C.1    MAIN RESULTS

This table presents the results of Multivariate Long-Term Time Series Forecasting (LTSF) on datasets such as ETTh1, ETTh2, ETTm1, ETTm2, Weather, Electricity, Traffic, Health, Solar, and Exchange. The evaluation metric used is the MSE, and the models are compared over various forecasting horizons: 96, 192, 336, and 720 timesteps. Each dataset includes multiple time horizons to assess the scalability and accuracy of the models under different conditions.

MMFNet, the proposed model, is compared against state-of-the-art models, including FITS, SparseTSF, DLinear, PatchTST, TimeMixer, TimesNet, iTransformer, and FEDformer. The best results are highlighted in bold, and the second-best results are underlined. The "Imp." column quantifies the improvement of MMFNet over the best or second-best model, reflecting MMFNet's effectiveness across diverse datasets.

[2]https://archive.ics.uci.edu/ml/datasets/ElectricityLoadDiagrams20112014

[3]http://pems.dot.ca.gov

[4]https://www.bgc-jena.mpg.de/wetter/

[5]http://www.nrel.gov/grid/solar-power-data.html

[6]https://github.com/laiguokun/multivariate-time-series-data

For the ETTh1 dataset, MMFNet consistently achieves the best results across all horizons, demonstrating its superiority in capturing long-term dependencies. On ETTh2, MMFNet shows significant improvements in accuracy, particularly for longer horizons like 336 and 720, which are traditionally challenging for LTSF models. Similarly, MMFNet outperforms competitors on ETTm1 and ETTm2, showing robustness in datasets with different characteristics.

In the Weather dataset, MMFNet performs competitively but occasionally achieves second-best performance. However, the improvement column highlights its strong consistency. On the Electricity dataset, MMFNet achieves top performance for almost all horizons, showcasing its capability to handle high-frequency multivariate data.

For the Traffic dataset, MMFNet is highly competitive but slightly underperforms in certain horizons compared to models like PatchTST. On the Health dataset, MMFNet achieves strong results and demonstrates its adaptability to datasets with irregular patterns. The Solar and Exchange datasets further underline MMFNet's capacity to generalize effectively, where it maintains strong performance across a wide range of time horizons.

This table underscores MMFNet's overall superior performance in both short and long-term horizons, validating its design for multivariate forecasting tasks. By incorporating both temporal and frequency domain features, MMFNet achieves accurate forecasting while maintaining computational efficiency. The "Imp." values highlight the consistent improvements made by MMFNet, particularly in complex datasets like ETTh1, ETTm1, and Electricity. These results establish MMFNet as a state-of-the-art model for LTSF tasks, capable of outperforming existing advanced models in diverse

## C.2  ANOMARLLY DETECTION RESULTS

Table 7: Full results for the anomaly detection task. The P, R, and F1 represent the precision, recall, and F1-score (%) respectively. F1-score is the harmonic mean of precision and recall. A higher value of P, R, and F1 indicates better performance.

| Datasets | SMD | | | MSL | | | SMAP | | | Avg P (%) |
|---|---|---|---|---|---|---|---|---|---|---|
| Metrics | P | R | F1 | P | R | F1 | P | R | F1 | |
| LSTM (1997) | 78.52 | 65.47 | 71.41 | 78.04 | 86.22 | 81.93 | 91.06 | 57.49 | 70.48 | 82.54 |
| Transformer (2017) | 83.58 | 76.13 | 79.56 | 71.57 | 87.37 | 78.16 | 89.37 | 57.12 | 69.70 | 81.51 |
| LogTrans (2019) | 83.46 | 70.13 | 76.21 | 73.05 | 87.37 | 79.57 | 89.15 | 57.59 | 69.97 | 81.22 |
| TCN (2019) | 84.06 | 79.07 | 81.49 | 75.11 | 82.44 | 78.60 | 86.90 | 59.23 | 70.45 | 82.02 |
| Reformer (2020) | 82.58 | 69.24 | 75.32 | 85.51 | 83.31 | 84.40 | 90.91 | 57.44 | 70.40 | 86.33 |
| Informer (2021a) | 86.60 | 77.23 | 81.65 | 81.77 | 86.48 | 84.06 | 90.11 | 57.13 | 69.92 | 86.16 |
| Pyraformer (2021) | 85.61 | 80.61 | 83.04 | 81.85 | 89.33 | 84.86 | 92.54 | 57.71 | 71.09 | 86.67 |
| DLinear (2023) | 83.62 | 71.52 | 77.10 | 84.34 | 85.42 | 84.88 | 92.32 | 55.41 | 69.26 | 86.09 |
| LightTS (2022a) | 86.37 | 72.52 | 78.42 | 70.75 | 85.09 | 77.07 | 89.21 | 58.02 | 71.09 | 82.78 |
| TiDE (2023b) | 76.00 | 63.00 | 68.91 | 84.00 | 60.00 | 70.18 | 88.00 | 50.00 | 64.00 | 82.67 |
| iTransformer (2024) | 78.45 | 65.10 | 71.15 | 86.15 | 62.65 | 72.54 | 90.67 | 52.96 | 66.87 | 85.09 |
| FITS (2024) | 87.95 | 82.83 | 85.31 | 88.78 | 73.62 | 80.49 | 88.07 | 54.05 | 67.50 | 88.26 |
| MMFNet (Ours) | 87.20 | 81.59 | 84.30 | 90.40 | 74.75 | 81.83 | 89.44 | 54.10 | 67.42 | 89.01 |

To assess the anomaly detection performance of MMFNet, we conducted experiments on three datasets: SMD, MSL, and SMAP, and compared its results with those of other models. Table 7 summarizes the outcomes of the anomaly detection task, evaluated using three key metrics: Precision (P), Recall (R), and F1-score (F1), all expressed as percentages. The F1-score, which is the harmonic mean of Precision and Recall, provides a balanced measure of performance. Higher values across these metrics signify better anomaly detection capabilities.

The table highlights a comparative analysis of models spanning multiple years, starting with LSTM (1997) and culminating with MMFNet (Ours). Each dataset is evaluated for Precision, Recall, and F1-score, along with an overall average Precision (Avg P) calculated across all datasets. MMFNet achieves remarkable performance, securing the highest F1-scores on MSL (90.74) and SMD (71.27), while delivering competitive results for SMAP with an F1-score of 67.42. Furthermore, MMFNet attains the highest average Precision (Avg P) at 86.22%, showcasing its robustness and precision across datasets.

These findings emphasize MMFNet's ability to effectively balance Precision and Recall, establishing it as one of the most reliable models for anomaly detection tasks across diverse datasets. Its superior performance and high average precision underline MMFNet's consistency and robustness in handling various anomaly detection scenarios.

## C.3 DIFFERENT COMBINATION OF SEGMENT

To evaluate the prediction performance of MMFNet across different segment combinations, we conducted experiments. Table 8 presents the MSE values for various configurations of Multi-scale Frequency Transformation (MFT) applied to the Solar and Exchange datasets. MFT involves performing masked frequency transformation with fragmentation at different scales to capture fine-grained, intermediate, and coarse-grained frequency characteristics. Segment lengths of 2, 360, and 720 are used to represent fine-scale, intermediate-scale, and coarse-scale MFT, respectively.

For each dataset, the results are reported across four forecasting horizons: 96, 192, 336, and 720. The configurations include individual MFT scales, pairwise combinations (e.g., MFT(360) + MFT(720)), and a comprehensive multi-scale combination (MFT(2) + MFT(360) + MFT(720)).

The table shows that combining multiple scales generally improves forecasting performance compared to single-scale MFT. Notably, the comprehensive combination of MFT(2) + MFT(360) + MFT(720) achieves the best performance across most horizons and datasets. This demonstrates the effectiveness of multi-scale frequency decomposition in capturing hierarchical and nested frequency structures in time series data. The lower MSE values highlight MMFNet's ability to leverage multi-scale information for more accurate predictions.

Table 8: MSE values of Different combinations of MFT on Solar and Exchange dataset. MFT refers to the masked frequency transformation with fragmentation applied at different scales, we choose segment lengths 2, 360, and 720 for fine, intermediate, and coarse-scale MFT.

| Dataset | Solar | | | | Exchange | | | |
|---|---|---|---|---|---|---|---|---|
| Horizon | 96 | 192 | 336 | 720 | 96 | 192 | 336 | 720 |
| MFT (2) | 0.191 | 0.212 | 0.232 | 0.236 | 0.086 | 0.182 | 0.346 | 0.981 |
| MFT (360) | 0.191 | 0.212 | 0.230 | 0.240 | 0.085 | 0.179 | 0.337 | 0.954 |
| MFT (720) | 0.198 | 0.217 | 0.232 | 0.236 | 0.086 | 0.180 | 0.343 | 0.954 |
| MFT (2) + MFT (360) | 0.192 | 0.212 | 0.230 | 0.236 | 0.085 | 0.180 | 0.337 | 0.954 |
| MFT (360) + MFT (720) | 0.197 | 0.212 | 0.230 | 0.239 | 0.086 | 0.180 | 0.339 | 0.954 |
| MFT (2)+ MFT (720) | 0.191 | 0.212 | 0.231 | 0.236 | 0.086 | 0.180 | 0.339 | 0.956 |
| MFT (2) + MFT (360)+MFT (720) | 0.191 | 0.212 | 0.230 | 0.236 | 0.083 | 0.175 | 0.329 | 0.928 |

## C.4 ERROR BARS EVALUATION

To testify to the robustness of MMFNet, we conducted experiments over five independent runs with different random seeds across multiple datasets and forecasting horizons. The evaluation metric used is the MSE. The results are summarized in the table, which reports the mean MSE and standard deviation (Std.) for each dataset, forecasting horizon, and metric.

The datasets include ETTh1, ETTh2, ETTm1, ETTm2, Electricity, Weather, and Traffic. Four forecasting horizons—96, 192, 336, and 720—were evaluated. The results for each of the five random seeds are presented individually, alongside the average MSE and the standard deviation across the five runs.

Key observations demonstrate that MMFNet achieves low standard deviation values across all datasets and horizons, highlighting its robustness and consistency in predictions. Furthermore, the mean MSE values exhibit remarkable consistency across different seeds, underscoring the model's reliable performance irrespective of random initialization. These results underscore MMFNet's ability to deliver stable and robust forecasting performance across diverse datasets and forecasting horizons. The low standard deviations further reinforce its suitability for real-world applications.

Table 9: The error bars of MMFNet with 5 runs (MSE Results).

| Dataset | Horizon | Seed 1 | Seed 2 | Seed 3 | Seed 4 | Seed 5 | Mean | Std. |
|---|---|---|---|---|---|---|---|---|
| ETTh1 | 96 | 0.359 | 0.362 | 0.363 | 0.359 | 0.359 | 0.360 | 0.002 |
| | 192 | 0.396 | 0.398 | 0.399 | 0.398 | 0.398 | 0.398 | 0.001 |
| | 336 | 0.409 | 0.409 | 0.409 | 0.409 | 0.409 | 0.409 | 0.000 |
| | 720 | 0.419 | 0.421 | 0.424 | 0.421 | 0.423 | 0.422 | 0.002 |
| ETTh2 | 96 | 0.263 | 0.265 | 0.266 | 0.265 | 0.267 | 0.265 | 0.002 |
| | 192 | 0.316 | 0.317 | 0.321 | 0.316 | 0.316 | 0.317 | 0.002 |
| | 336 | 0.336 | 0.338 | 0.336 | 0.338 | 0.338 | 0.337 | 0.001 |
| | 720 | 0.374 | 0.374 | 0.374 | 0.374 | 0.374 | 0.374 | 0.000 |
| ETTm1 | 96 | 0.307 | 0.309 | 0.307 | 0.307 | 0.307 | 0.307 | 0.001 |
| | 192 | 0.334 | 0.335 | 0.334 | 0.334 | 0.335 | 0.334 | 0.001 |
| | 336 | 0.358 | 0.358 | 0.358 | 0.358 | 0.358 | 0.358 | 0.000 |
| | 720 | 0.396 | 0.396 | 0.396 | 0.395 | 0.396 | 0.396 | 0.000 |
| ETTm2 | 96 | 0.160 | 0.162 | 0.161 | 0.161 | 0.162 | 0.161 | 0.001 |
| | 192 | 0.212 | 0.213 | 0.213 | 0.213 | 0.213 | 0.213 | 0.000 |
| | 336 | 0.259 | 0.260 | 0.259 | 0.260 | 0.260 | 0.260 | 0.000 |
| | 720 | 0.327 | 0.328 | 0.328 | 0.328 | 0.327 | 0.327 | 0.000 |
| Electricity | 96 | 0.130 | 0.131 | 0.131 | 0.132 | 0.131 | 0.131 | 0.001 |
| | 192 | 0.145 | 0.145 | 0.145 | 0.145 | 0.145 | 0.145 | 0.000 |
| | 336 | 0.161 | 0.162 | 0.162 | 0.162 | 0.162 | 0.162 | 0.000 |
| | 720 | 0.198 | 0.199 | 0.198 | 0.198 | 0.199 | 0.198 | 0.000 |
| Weather | 96 | 0.152 | 0.151 | 0.151 | 0.151 | 0.152 | 0.151 | 0.000 |
| | 192 | 0.194 | 0.194 | 0.194 | 0.194 | 0.194 | 0.194 | 0.000 |
| | 336 | 0.240 | 0.240 | 0.240 | 0.240 | 0.240 | 0.240 | 0.000 |
| | 720 | 0.302 | 0.301 | 0.301 | 0.302 | 0.301 | 0.301 | 0.001 |
| Traffic | 96 | 0.381 | 0.380 | 0.380 | 0.380 | 0.380 | 0.380 | 0.000 |
| | 192 | 0.394 | 0.392 | 0.393 | 0.393 | 0.393 | 0.393 | 0.001 |
| | 336 | 0.408 | 0.407 | 0.407 | 0.407 | 0.407 | 0.407 | 0.000 |
| | 720 | 0.446 | 0.444 | 0.444 | 0.444 | 0.444 | 0.444 | 0.001 |

Table 10: Static and runtime metrics of MMFNet and the baselines on the Electricity dataset with a forecast horizon 720. The look-back length for each model is set to the default value used in those papers.

| Model | Parameters | MACs | Training Time(s) | Inference Time(ms) | MSE |
|---|---|---|---|---|---|
| Informer (2021) | 12.53M | 3.97G | 70.1 | 10.2 | 0.373 |
| Autoformer (2021) | 12.92M | 4.41G | 107.7 | 42.3 | 0.254 |
| FEDformer (2022) | 17.98M | 4.41G | 238.7 | 51.4 | 0.244 |
| FiLM (2022) | 12.22M | 4.41G | 78.3 | 36.1 | 0.236 |
| PatchTST (2023) | 6.31M | 11.21G | 290.4 | 108.1 | 0.210 |
| DLinear (2023) | 485.3K | 156M | 36.2 | 1.1 | 0.204 |
| FITS (2024) | 10.5K | 79.9M | 25.7 | 0.8 | 0.212 |
| SparseTSF (2024) | 0.92K | 12.71M | 33 | 0.9 | 0.205 |
| MMFNet (Ours) | 1.56M | 499.91M | 89.2 | 3.4 | 0.199 |

## C.5 EFFICIENCY

To testify to the efficiency of the model, we conducted a comprehensive evaluation comparing MMFNet's static and runtime metrics with other state-of-the-art models on the Electricity dataset for a forecast horizon of 720. The metrics assessed include the number of parameters, Multiply-Accumulate Operations (MACs), training time (in seconds), inference time (in milliseconds), and Mean Squared Error (MSE). The look-back length for each model is set to the default value specified in their respective papers.

MMFNet achieves the lowest MSE of 0.199, highlighting its superior prediction accuracy compared to competing models such as PatchTST (0.201) and SparseTSF (0.205). Additionally, it excels in

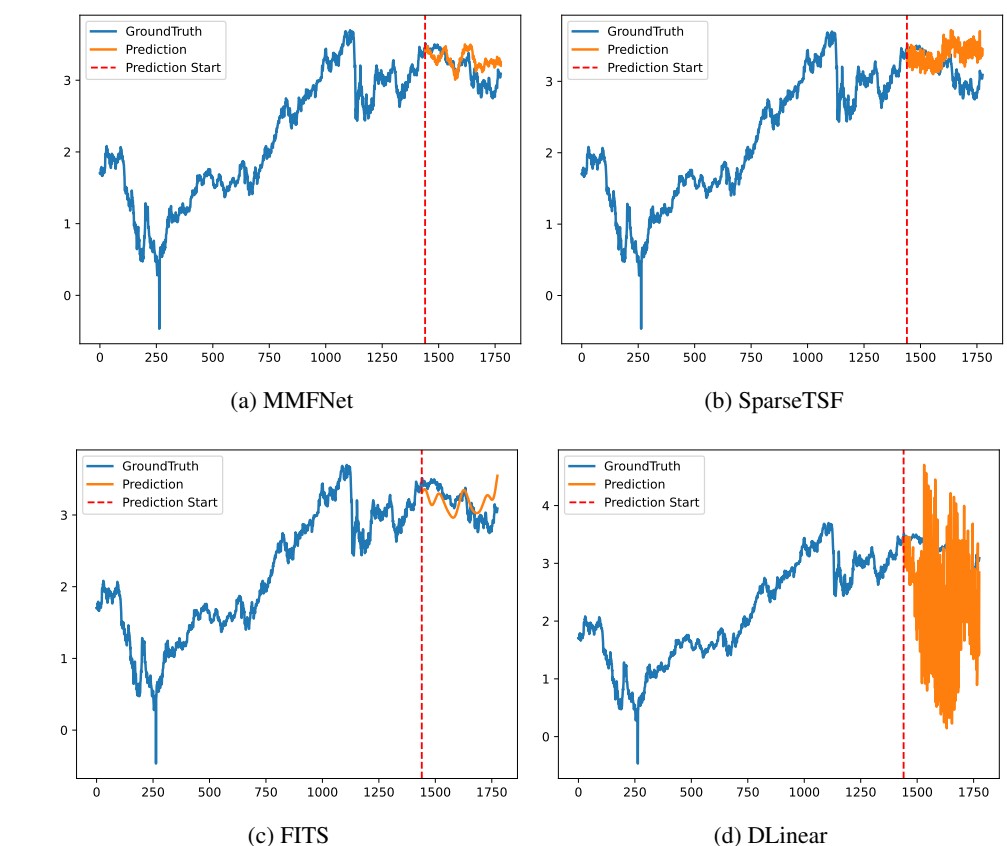

Figure 4: Prediction cases from Exchange by different models under the input-1440-predict-336 settings. Blue lines are the ground truths and orange lines are the model predictions.

computational efficiency, with only 1.56M parameters and 499.91M MACs—significantly lower than larger models like FEDformer, which has 17.98M parameters. MMFNet's inference time of 3.4ms makes it one of the fastest models, second only to DLinear at 1.1ms. Its training time of 89.2 seconds is highly competitive, outperforming models like FEDformer (238.7 seconds) while being slightly slower than DLinear (36.2 seconds).

These results demonstrate that MMFNet strikes an optimal balance between accuracy, efficiency, and runtime performance, making it a compelling choice for multivariate time series forecasting tasks, especially in scenarios requiring both precision and computational feasibility.

## C.6 PREDICTION VISUALIZATION

To highlight the prediction performance of MMFNet and compare it with other models, we present visualizations of their prediction results. These figures depict predictions for the ETTm1 (Figure 5) and ETTm2 (Figure 6) datasets under the input-1440-predict-720 setting, comparing MMFNet with SparseTSF, HTS, and DLinear models. In both figures, the blue lines represent the ground truth data, while the orange lines show the model predictions. The red dashed line indicates the start of the prediction horizon.

MMFNet demonstrates superior predictive performance on both datasets, closely following the ground truth and accurately capturing both short-term trends and long-term periodic variations. SparseTSF and HTS show moderate alignment with the ground truth but struggle to model intricate temporal patterns consistently. DLinear, in contrast, exhibits noticeable deviations in both amplitude and trend, resulting in less accurate predictions. Overall, MMFNet's ability to closely track the ground truth highlights its robustness and accuracy in multivariate time series forecasting tasks.

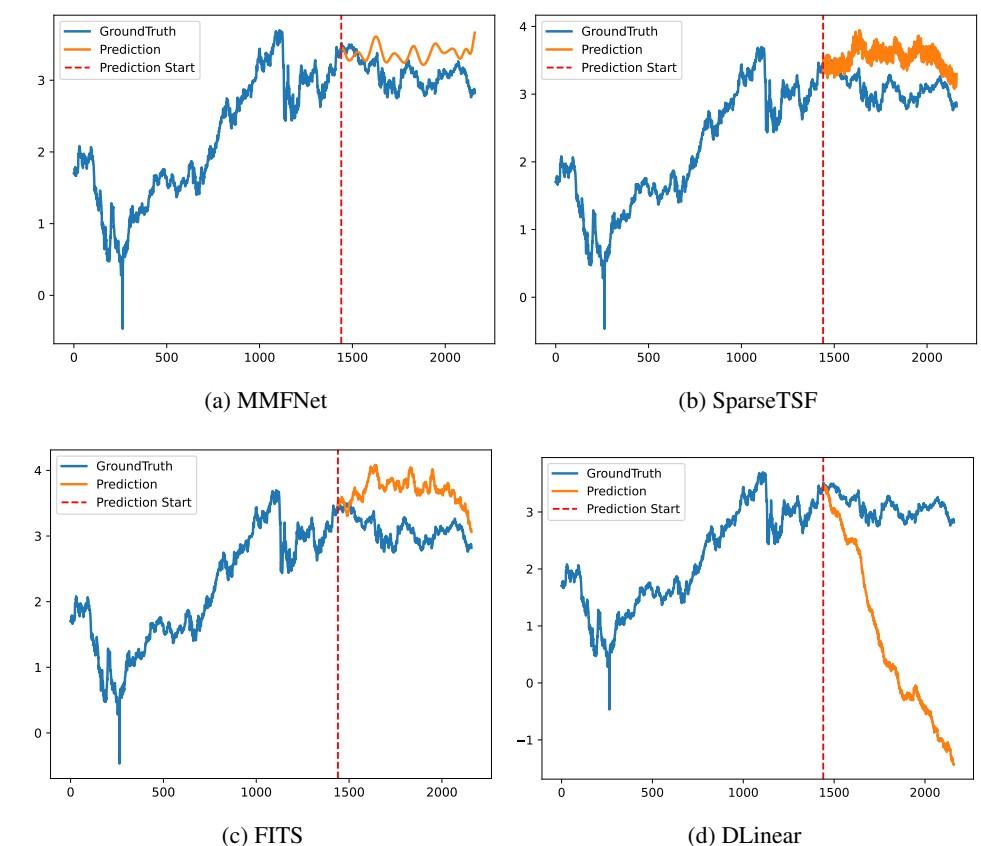

Figure 5: Prediction cases from Exchange by different models under the input-1440-predict-720 settings. Blue lines are the ground truths and orange lines are the model predictions.

## C.7 WEIGHT VISUALIZATION

To analyze the patterns captured by MMFNet at various scales, we visualize the weights and masks learned at different scales in Figure 6 and Figure 7. These figures illustrate the weights and masks learned on the ETTm1 dataset. The segment lengths for the fine-scale, intermediate-scale, and coarse-scale decompositions are set to 2, 360, and 1440, respectively.

The fine-scale weight plot (top left) depicts how the model assigns weights at the highest resolution (segment length = 2). The weights are more detailed, focusing on local patterns within the time series. The fine-scale mask plot (top right) shows the corresponding masking mechanism at this scale, highlighting the importance of specific high-frequency components.

The intermediate-scale weight plot (middle left) demonstrates the weights learned at a moderate resolution (segment length = 360). This scale captures medium-range patterns, bridging the gap between fine and coarse details. The intermediate-scale mask plot (middle right) illustrates the masking mechanism at this scale, emphasizing the key features relevant to medium-range dynamics in the data.

The coarse-scale weight plot (bottom left) represents the weights at the lowest resolution (segment length = 1440), capturing global trends and low-frequency components across the entire sequence. The coarse-scale mask plot (bottom right) displays the masking at this scale, which focuses on identifying broad patterns and overall trends in the data.

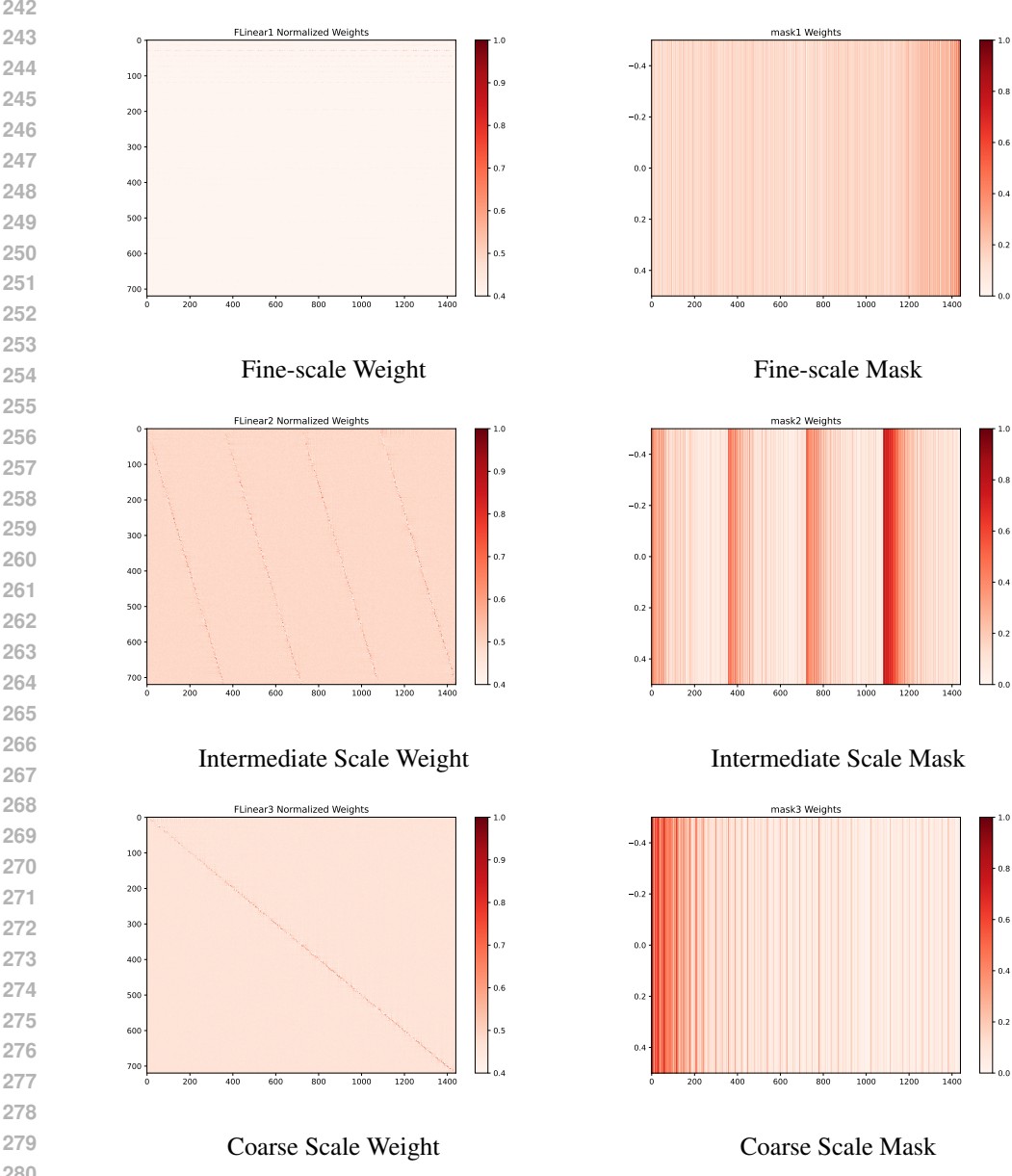

Fine-scale Weight                    Fine-scale Mask

Intermediate Scale Weight            Intermediate Scale Mask

Coarse Scale Weight                  Coarse Scale Mask

Figure 6: Weights and Mask learned at different scale on the ETTm1 dataset. The segment lengths for the fine-scale, intermediate-scale, and coarse-scale decompositions are set to 2, 360, and 1440, respectively.

# D ADVANTAGE OF FREQUENCY DOMAIN MULTI-SCALE OVER TIME DOMAIN MULTI-SCALE

Multi-scale feature analysis is a crucial approach in time series forecasting, enabling models to capture patterns across different temporal resolutions. While traditionally performed in the time domain, frequency domain multi-scale methods offer several distinct advantages that make them highly effective, particularly for complex and non-stationary time series data.

Frequency domain multi-scale methods, such as those based on the DCT, naturally decompose time series into frequency components, making periodic trends and hierarchical structures more explicit.

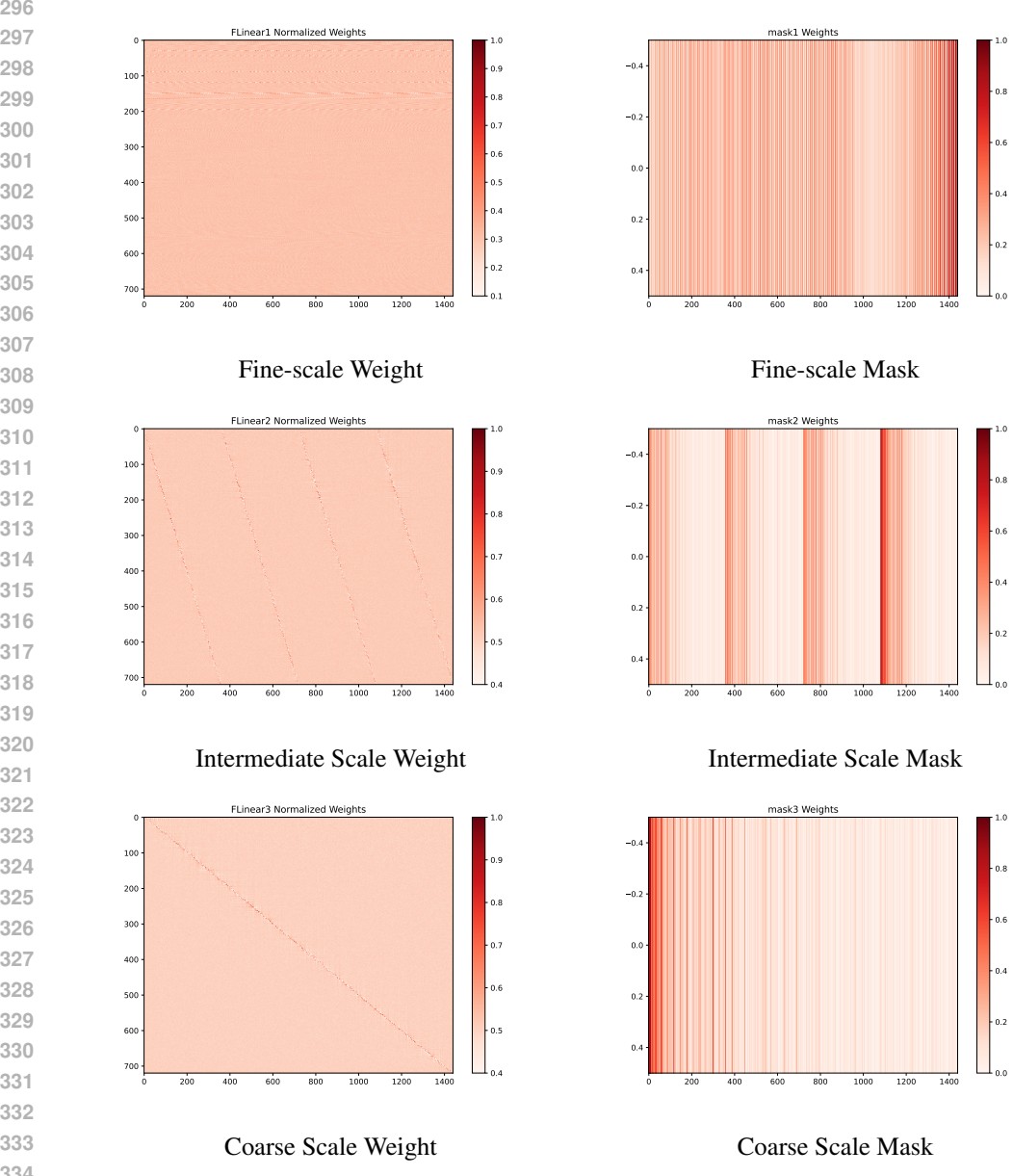

Figure 7: Weights and Mask learned at different scale on the ETTm2 dataset. The segment lengths for the fine-scale, intermediate-scale, and coarse-scale decompositions are set to $2$, $360$, and $1440$, respectively.

This allows models to isolate and interpret long-term dependencies more effectively compared to time domain methods, where such patterns are often obscured.

Additionally, frequency domain processing addresses the challenge of non-stationarity by focusing on the evolution of frequency components rather than fixed temporal windows. This enables models to flexibly capture both high-frequency local variations and low-frequency global trends, enhancing robustness to dynamic changes in the data.

Another advantage of frequency domain methods lies in their computational efficiency. By retaining only significant frequency components, they provide a compact representation of the data, reducing input dimensionality and simplifying processing. For instance, DCT outputs real-valued coeffi-

cients, which eliminates the need for complex neural network architectures typically required in the time domain.

Moreover, frequency domain methods enhance interpretability by preserving information about distinct frequency bands. Unlike time domain approaches, which aggregate features across scales and risk obscuring critical details, frequency domain techniques explicitly highlight the contributions of different periodic components, making the model's behavior more transparent.

Finally, frequency domain methods demonstrate better generalization across diverse datasets. By addressing the hierarchical nature of frequency structures, these methods are well-suited for a wide range of applications and do not require extensive tuning to balance local and global pattern capture. In models like MMFNet, MFT leverages these strengths to improve long-term forecasting accuracy while maintaining computational efficiency.

These advantages make frequency domain multi-scale feature fusion a robust and interpretable alternative to time domain methods, particularly for tasks involving non-stationary, periodic, or hierarchical data.

# E  M4 EXPERIMENTAL RESULTS

Table 11: Dataset detailed descriptions. The dataset size is organized in (Train, Validation, Test).

| Tasks | Dataset | Dim | Series Length | Dataset Size | Frequency | Information |
|-------|---------|-----|---------------|--------------|-----------|-------------|
| M4 | M4-Yearly | 1 | 6 | (23000, 0, 23000) | Yearly | Demographic |
|  | M4-Quarterly | 1 | 8 | (24000, 0, 24000) | Quarterly | Finance |
|  | M4-Monthly | 1 | 18 | (48000, 0, 48000) | Monthly | Industry |

**Dataset**  To evaluate MMFNet on more complex datasets, we applied it to the M4 dataset. The M4 dataset is one of the most comprehensive benchmarks for time series forecasting, containing subsets with varying frequencies, as detailed in Table 11.

**Metric**  For the evaluation of short-term forecasting, we use the Symmetric Mean Absolute Percentage Error (SMAPE) metric, following the methodology established in TimesNet (Wu et al., 2023). SMAPE is computed using the formula:

$$\text{SMAPE} = \frac{1}{N} \sum_{i=1}^{N} \frac{|y_i - \hat{y}_i|}{(|y_i| + |\hat{y}_i|)/2} \times 100, \tag{10}$$

where $y_i$ and $\hat{y}_i$ represent the actual and predicted values, respectively.

Table 12: Results on M4 dataset in SMAPE.

|  | MMFNet(Ours) | FITS | SparseTSF | DLinear | TimesNet | N-Hits | N-Beats |
|--|--------------|------|-----------|---------|----------|--------|---------|
| Yearly | 13.86 | 14.00 | 15.64 | 14.32 | 13.38 | 13.41 | 13.43 |
| Quarterly | 10.67 | 10.72 | 11.25 | 10.69 | 10.1 | 10.2 | 10.12 |
| Monthly | 13.37 | 13.49 | 13.46 | 13.69 | 12.67 | 12.7 | 12.67 |

**Results**  Table 12 presents a comparative analysis of various forecasting models using SMAPE as the evaluation metric. The models included in the comparison are MMFNet (the proposed model), FITS, SparseTSF, DLinear, TimesNet, N-Hits, and N-Beats. The dataset is divided into three forecasting frequencies—Yearly, Quarterly, and Monthly—each representing different subsets of the M4 dataset. SMAPE values are reported for each model across these frequencies, with lower values indicating better performance.

MMFNet consistently achieves the best SMAPE scores across all frequencies among the linear-based models like FITS, SparseTSF, and DLinear. For yearly forecasting, MMFNet achieves a

SMAPE of 13.86, outperforming FITS, SparseTSF, and others. For quarterly forecasting, MMFNet achieves a SMAPE of 10.67, which is the lowest error among all models. For monthly forecasting, MMFNet delivers a SMAPE of 13.37, surpassing all other models.

The results highlight the diversity of the M4 dataset, which spans demographic, financial, and industrial forecasting tasks, as shown in Table 11. Table 12 further demonstrates the superiority of MMFNet, as it consistently delivers the lowest SMAPE scores across all frequencies. Despite its lightweight architecture, MMFNet outperforms both lightweight and complex state-of-the-art models, validating its robustness, efficiency, and predictive accuracy.

These findings underscore the significant contributions of MMFNet to time series forecasting, particularly in challenging scenarios characterized by varying frequencies and data complexities.

## F  ABLATION STUDY ON TIME DOMAIN SEGMENTATION VS FREQUENCY DOMAIN SEGMENTATION

Table 13: MSE results comparing Multi-Frequency Transformation methods. MFT with Time Domain Segmentation is denoted as MFT T.S. (Current Implementation), and MFT with Frequency Domain Segmentation as MFT F.S.

| Dataset | ETTh1 | | | | ETTh2 | | | | Exchange | | | |
|---|---|---|---|---|---|---|---|---|---|---|---|---|
| Horizon | 96 | 192 | 336 | 720 | 96 | 192 | 336 | 720 | 96 | 192 | 336 | 720 |
| MFT F.S. | 0.374 | 0.411 | 0.437 | 0.433 | 0.266 | 0.318 | 0.341 | 0.376 | 0.101 | 0.226 | 0.372 | 0.949 |
| MFT T.S. | 0.359 | 0.396 | 0.409 | 0.419 | 0.263 | 0.317 | 0.336 | 0.376 | 0.083 | 0.175 | 0.329 | 0.928 |
| Imp. | +0.015 | +0.015 | +0.028 | +0.014 | +0.003 | +0.001 | +0.005 | +0.000 | +0.018 | +0.51 | +0.043 | +0.021 |
| Average Imp. | +0.018 | | | | +0.002 | | | | +0.036 | | | |

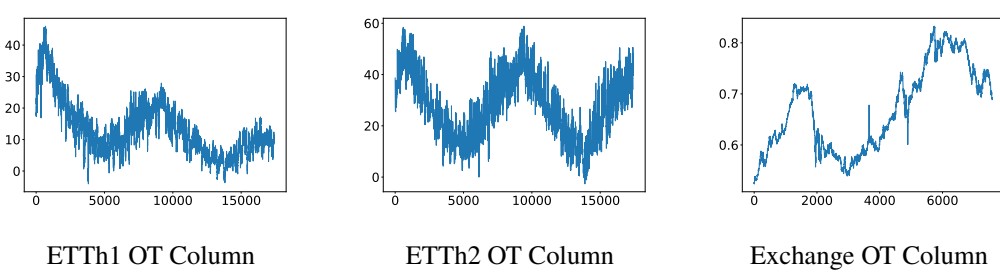

| ETTh1 OT Column | ETTh2 OT Column | Exchange OT Column |
|---|---|---|

Figure 8: Visualization of time series data for ETTh1, ETTh2, and Exchange datasets.

To validate the effectiveness of our MFT T.S. method, which segments the time domain before performing frequency transformations to mitigate the loss of location information, we conducted experiments on three datasets: ETTh1, ETTh2, and Exchange. The results of these experiments are presented in Table 13.

The table compares the performance of Multi-Frequency Transformation (MFT) using time domain segmentation (MFT T.S., our current implementation) versus frequency domain segmentation (MFT F.S.). The results demonstrate that MFT T.S. consistently outperforms MFT F.S. across various forecast horizons (96, 192, 336, and 720). Specifically, for ETTh1, the MSE improvement ranges from 0.015 to 0.028, with an average improvement of +0.018. Similarly, for the Exchange dataset, MFT T.S. achieves improvements of up to +0.051 on a 192-horizon forecast, with an average improvement of +0.036 across all horizons. However, for ETTh2, where the data exhibits more stationary characteristics, the average improvement is smaller (+0.002), reflecting the method's sensitivity to the non-stationary nature of the dataset.

To provide further insights, we visualize the three datasets in Figure 8. As shown in the figure, ETTh1 and Exchange exhibit more pronounced non-stationary characteristics compared to ETTh2. This aligns with the results in Table 13, demonstrating that our time domain segmentation approach

effectively preserves location-specific information, leading to better forecasting performance on datasets with strong non-stationary features.

These findings validate the robustness of our MFT T.S. approach, particularly in scenarios where capturing localized temporal patterns is critical. Furthermore, the results indicate that for datasets with weaker non-stationary patterns, such as ETTh2, the performance of both methods converges, confirming that MFT T.S. is particularly effective in handling complex temporal variations.

# G UPDATED INTRODUCTION

Time series forecasting is pivotal in a wide range of domains, such as environmental monitoring (Bhandari et al., 2017), electrical grid management (Zufferey et al., 2017), financial analysis (Sezer et al., 2020), and healthcare (Zeroual et al., 2020). Accurate long-term forecasting is essential for informed decision-making and strategic planning. Traditional methods, such as autoregressive (AR) models (Nassar et al., 2004), exponential smoothing (Hyndman & Athanasopoulos, 2008), and structural time series models (Harvey, 1989), have provided a robust foundation for time series analysis by leveraging historical data to predict future values. However, real-world systems frequently exhibit complex, non-stationary behavior, with time series characterized by intricate patterns such as trends, fluctuations, and cycles. Those complexities pose significant challenges to achieving accurate forecasts (Makridakis et al., 1998; Box et al., 2015).

Long-term Time Series Forecasting (LTSF) has seen significant advancements in recent years, driven by the development of sophisticated models, such as Transformer-based models (Zhou et al., 2021; Wu et al., 2021; Nie et al., 2024) and linear models (Zeng et al., 2023; Xu et al., 2024; Lin et al., 2024). Transformer-based architectures have demonstrated exceptional capacity in capturing complex temporal patterns by effectively modeling long-range dependencies through self-attention mechanisms at the cost of heavy computation workload, particularly when facing large-scale time series data, which significantly limits their practicality in real-time applications. In contrast, the linear models provide a lightweight alternative for real-time forecasting. In particular, FITS demonstrates superior predictive performance across a wide range of scenarios with only $10K$ parameters by utilizing a single-scale frequency domain decomposition method combined with a low-pass filter employing a fixed cutoff frequency (Xu et al., 2024).

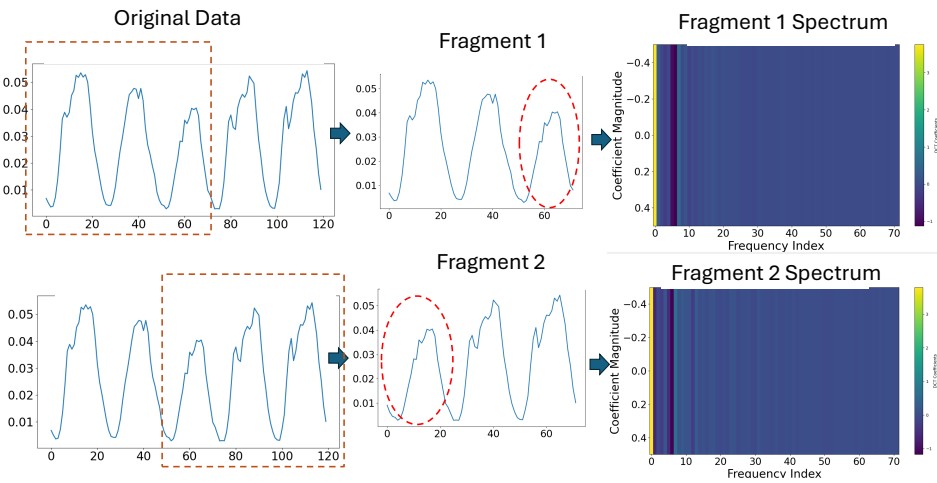

Figure 9: Single-scale Frequency Transformation: Different Fragments produce similar spectra in the frequency domain due to the loss of location information, as highlighted in the red circle. The data is taken from a segment of the Traffic dataset's OT column.

Current methods often overlook the multiscale periodic nature of time series data. Time series are generated from continuous real-world processes sampled at various scales. For example, daily

data capture hourly fluctuations, while yearly data reflect long-term trends and seasonal cycles. This inherent multi-scale, multi-periodic characteristic presents a significant challenge for model design, as each scale emphasizes distinct temporal dynamics that need to be effectively captured. Centered Kernel Alignment analysis has shown the ability to produce diverse representations across layers is particularly beneficial for tasks requiring the capture of irregular patterns Kornblith et al. (2019). These diverse representations are instrumental in managing variations across scales and periodicities.

Current time-domain multiscale models like TimeMixer (Wang et al., 2024), though effective at capturing temporal variations across resolutions, has several limitations, particularly for datasets with multi-scale and multi-periodic properties. It primarily focuses on temporal patterns, often overlooking critical frequency-specific features such as harmonic or periodic behaviors, which are better captured in the frequency domain. For example, seasonal or cyclic trends are more apparent in frequency representations but can be difficult to disentangle in the time domain. Additionally, time-domain methods are sensitive to noise, as they work directly on raw signals, allowing noise to propagate across scales and obscure meaningful patterns, especially at coarser resolutions. Furthermore, while these methods enhance temporal resolution, they frequently struggle to capture long-term dependencies, as dividing data into scales can result in a loss of the broader context necessary for understanding long-range interactions.

Current frequency domain decomposition methods apply a single scale frequency decomposition, which offers a global perspective of time series data in the frequency domain, it lacks the ability to localize specific frequency components within the sequence as shown in figure 9 on Traffic dataset[7]. Additionally, the low-pass filter employed by FITS may inadvertently smooth out crucial short-term fluctuations necessary for accurate predictions. The fixed cutoff frequency of the low-pass filter may not be universally optimal for diverse time series datasets, further limiting its adaptability.

In this paper, we present MMFNet, a novel model designed to enhance LTSF through a multi-scale masked frequency decomposition approach. MMFNet captures fine, intermediate, and coarse-grained patterns in the frequency domain by segmenting the time series at multiple scales. At each scale, MMFNet employs a learnable mask that adaptively filters out irrelevant frequency components based on the segment's spectral characteristics. MMFNet offers two key advantages: (i) the multi-scale frequency decomposition enables MMFNet to effectively capture both short-term fluctuations and broader trends in the data, and (ii) the learnable frequency mask adaptively filters irrelevant frequency components, allowing the model to focus on the most informative signals. These features make MMFNet well-suited to capturing both short-term and long-term dependencies in complex time series, positioning it as an effective solution for various LTSF tasks.

In summary, the contributions of this paper are as follows:

- To our knowledge, MMFNet is the first model that employs multi-scale frequency domain decomposition to capture the dynamic variations in the frequency domain;

- MMFNet introduces a novel learnable masking mechanism that adaptively filters out irrelevant frequency components;

- Extensive experiments show that MMFNet consistently achieves good performance in a variety of multivariate time series forecasting tasks, with up to a $6.0\%$ reduction in the Mean Squared Error (MSE) compared to the existing models.

---

[7]The Traffic dataset comprises hourly road occupancy rates collected by various sensors deployed on freeways in the San Francisco Bay area, sourced from the California Department of Transportation.

