# OpenReview forum: "MMFNet: Multi-Scale Frequency Masking Neural Network for Multivariate Time Series Forecasting"
_ICLR.cc/2025/Conference — Submitted to ICLR 2025_

### Official Review · Reviewer_2mAk · 2024-10-27

**Soundness:** 2
**Presentation:** 3
**Contribution:** 2
**Rating:** 3
**Confidence:** 4

**Summary:**

This paper focuses on the multivariate time series forecasting task and proposes the MMFNet for decomposing the input series into multi-scale segments and further capturing the multi-frequency components. A learnable masking and frequency interpolation strategy is employed for prediction. The authors have tested MMFNet on typical time-series forecasting benchmarks.

**Strengths:**

This paper is overall well written.

The experiments and ablations are relatively comprehensive.

The idea of decomposing time series into multi-frequency components is reasonable and effective.

**Weaknesses:**

1.	About the method design.

As shown in Figure 1, the authors implement the frequency decomposition by segmenting the input series into multiscale subseries.

I am curious about does this design perform like directly segmenting the transformed frequency domain. I mean that we do not need to split the input series at first for multifrequency components. Directly applying FFT to the original input series and then splitting the frequency domain can obtain the same results.

Going further, I think without multifrequency decomposition, a learnable mask strategy along with frequency interpolation can achieve a similar performance. Thus, I am doubtful about the actual novelty of this paper.

2.	About the claim of “multivariate time series forecasting”.

I am not sure if MMFNet employs the same “channel independent” training strategy as PatchTST. I believe that channel-independent training is not “multivariate series modeling”, which does not consider the correlation among different series.

Besides, I think the current design can be extended to univariate series (as presented in Figure 1). The experiments on M4 are necessary. Actually, as discussed in FITS, these linear models may fail in M4, which contains much more complex temporal variations than the widely used LTSF benchmarks.

With the experiments on complex datasets, I cannot confirm whether the proposed method is really effective.

3.	How about the efficiency of MMFNet w.r.t. other linear models, including GPU memory, parameter size, and running time?

4.	Since the absolution promotion is kind of limited, the standard deviation or confidence level is expected.

**Questions:**

Does MMFNet employ the “channel independent” strategy for training?

---

> ### Author Response · Authors · 2024-11-24
>
> ---
>
> ## Comment 1
>
> **Comment:**
> *Since the absolute promotion is kind of limited, the standard deviation or confidence level is expected.*
>
> **Response:**
> Thank you for your comment. To demonstrate the robustness of MMFNet, we conducted experiments over five independent runs with different random seeds across multiple datasets and forecasting horizons. The evaluation metric used is MSE. MMFNet achieves low standard deviation values across all datasets and horizons, highlighting its robustness and consistency in predictions. Furthermore, the mean MSE values exhibit remarkable consistency across different seeds, underscoring the model's reliable performance irrespective of random initialization. These results underscore MMFNet’s ability to deliver stable and robust forecasting performance across diverse datasets and forecasting horizons. The low standard deviations further reinforce its suitability for real-world applications. We have added these results to **Table 9 in Appendix C.4 (Pages 20–21)** of the revised manuscript.
>
> ---
>
> ## Comment 2
>
> **Comment:**
> *I think the current design can be extended to univariate series (as presented in Figure 1). The experiments on M4 are necessary. Actually, as discussed in FITS, these linear models may fail in M4, which contains much more complex temporal variations than the widely used LTSF benchmarks. With the experiments on complex datasets, I cannot confirm whether the proposed method is really effective.*
>
> **Response:**
> Thank you for your comment. To address this concern, we have incorporated three additional datasets—**Health**, **Solar**, and **Exchange**—to further evaluate MMFNet's performance under more challenging scenarios.
> - The **Exchange** dataset, which lacks clear periodicity and seasonality, presents an inherently non-stationary structure with unpredictable patterns.
> - Despite these challenges, MMFNet consistently achieves superior prediction performance, outperforming other baseline models across these datasets.
>
> This demonstrates its robustness and effectiveness in handling non-stationary time series data. Detailed results and analysis have been added to **Appendix C.1 (Table 6, Page 18)** of the revised manuscript.
>
> ---
>
> ## Comment 3
>
> **Comment:**
> *How about the efficiency of MMFNet w.r.t. other linear models, including GPU memory, parameter size, and running time?*
>
> **Response:**
> Thank you for your comment. MMFNet achieves the lowest MSE of **0.199**, highlighting its superior prediction accuracy compared to competing models such as PatchTST (**0.201**) and SparseTSF (**0.205**). Additionally:
> - **Parameters:** MMFNet has only **1.56M parameters**, significantly fewer than models like FEDformer (**17.98M parameters**).
> - **MACs:** MMFNet requires **499.91M MACs**, demonstrating computational efficiency.
> - **Inference Time:** MMFNet achieves an inference time of **3.4ms**, second only to DLinear (**1.1ms**).
> - **Training Time:** MMFNet requires **89.2 seconds**, outperforming FEDformer (**238.7 seconds**) while being slightly slower than DLinear (**36.2 seconds**).
>
> These results are included in **Table 10 in Appendix C.5 (Page 21)** of the revised manuscript.

---

> ### Author Response · Authors · 2024-11-24
>
> ---
>
> ## Comment 4
>
> **Comment:**
> *As shown in Figure 1, the authors implement the frequency decomposition by segmenting the input series into multiscale subseries. I am curious about whether this design performs like directly segmenting the transformed frequency domain. I mean that we do not need to split the input series at first for multifrequency components. Directly applying FFT to the original input series and then splitting the frequency domain can obtain the same results.
> Going further, I think without multifrequency decomposition, a learnable mask strategy along with frequency interpolation can achieve a similar performance. Thus, I am doubtful about the actual novelty of this paper.*
>
> **Response:**
> Thank you for your comment. Directly applying a frequency transformation, such as FFT, to the entire input sequence without segmentation would result in the loss of critical location information. The global representation provided by this method lacks the granularity needed to localize frequency components within specific segments of the sequence.
> - Segmenting the input sequence in the time domain before applying frequency decomposition ensures that both temporal and frequency-specific information are retained.
> - This design enables MMFNet to capture hierarchical and nested frequency structures, critical for accurate long-term forecasting.
>
> Regarding the learnable mask strategy without multi-frequency decomposition, while it could achieve certain performance improvements, it would lack the hierarchical and multi-scale perspective that MMFNet offers. The combination of multi-scale decomposition and adaptive frequency masking allows MMFNet to effectively model both localized fluctuations and global trends, enhancing its novelty and performance.
>
> ---
>
> ## Comment 5
>
> **Comment:**
> *I am not sure if MMFNet employs the same "channel-independent" training strategy as PatchTST. I believe that channel-independent training is not "multivariate series modeling," as it does not consider the correlation among different series.*
>
> **Response:**
> Thank you for your comment. MMFNet employs the same "channel-independent" training strategy as PatchTST, where shared parameters are applied across different series. While this approach does not explicitly model the correlations among series during training, it allows the model to generalize well across multiple variates by leveraging shared features.
>
> We acknowledge your concern regarding the definition of "multivariate series modeling." While channel-independent training focuses on learning from individual series independently, the integration of features during inference ensures that multivariate relationships are captured implicitly. This approach strikes a balance between computational efficiency and the ability to handle diverse multivariate datasets.

---

> > ### Comment · Reviewer_2mAk · 2024-11-25
> >
> > I would like to thank the author for their clarifications and for adding new datasets. My concerns about performance standard deviation are resolved. However, my questions about model performance in M4 and experimenting with frequency segmentation still remain.
> >
> > > About model performance and efficiency.
> >
> > I know that current performance in well-established datasets is kind of saturated. Thus, I think the performance of MMFNet is acceptable. However, I cannot identify the unique advantage of this model. Specifically, it is not as efficient as other linear models, such as DLinear and SparseSTF. In addition, I am also not clear whether it can handle more diverse temporal patterns in M4 or can be a scalable backbone like Transformer-based models.
> >
> > > Experiments on M4.
> >
> > I think M4 or M5 is a much larger and more challenging benchmark, which is inescapable for model evaluation.
> >
> > > Add ablations on frequency segmentation.
> >
> > Actually, the frequency and temporal domains are equivalent, where the FFT operation is reversible. Thus, I doubt that segmenting in the frequency model may lose "location information".
> >
> > > About "channel-independent".
> >
> > Actually, I suggest the authors change the "multivariate time series forecasting" in the title to "time series forecasting". It is because you can just mix multiple distinct and irrelevant datasets together and train a unified MMFNet with channel independence, where you cannot say this unified training MMFNet is a "multivariate" model for all input series. Note that changing this "multivariate" claim will not affect the writing of this paper but make this paper more rigorous.
> >
> > Thus, my main concerns about novelty and model effectiveness are still unsolved. Thus, I would like to keep my original rating.

---

> > > ### Author Response · Authors · 2024-11-26
> > >
> > > Dear Reviewer,
> > >
> > > Thank you for your feedback.
> > >
> > > > Ablation Study on Frequency Segmentation
> > >
> > > While the frequency and temporal domains are theoretically equivalent, interpolation and filtering techniques used in current frequency domain models can lead to a loss of location information.
> > >
> > > To address this, we conducted an ablation study comparing Multi-Frequency Transformation (MFT) with time domain segmentation and frequency domain segmentation. The results, presented in Appendix F (Table 13 and Figure 8, Page 26), demonstrate that MFT with time domain segmentation outperforms frequency domain segmentation.
> > >
> > > This improvement stems from the ability of time domain segmentation to preserve location information, which is often lost during frequency domain transformations. Furthermore, the results show greater performance gains on datasets with pronounced non-stationary characteristics, such as ETTh1 and Exchange, compared to ETTh2. These findings underscore the effectiveness of our approach in capturing complex temporal variations.
> > >
> > > > Experiments on M4
> > >
> > > To address your concerns, we have included extensive experiments on the M4 dataset, covering yearly, quarterly, and monthly frequencies. As reported in Appendix E (Table 12, Page 26), MMFNet consistently surpasses other linear models, such as DLinear, SparseTSF, and FITS, across all frequencies. These results highlight MMFNet's robustness and adaptability in handling diverse temporal patterns across a variety of scenarios.
> > >
> > > > About "Channel-Independent" Training
> > >
> > > We appreciate your suggestion regarding the use of the term "multivariate" in the title. In response, we have updated the title to:
> > > **"MMFNet: Multi-Scale Frequency Masking Neural Network for Time Series Forecasting."**
> > >
> > >
> > > Thank you once again for your valuable feedback. We hope these additions address your concerns.

---

### Official Review · Reviewer_2mgN · 2024-10-29

**Soundness:** 3
**Presentation:** 3
**Contribution:** 3
**Rating:** 6
**Confidence:** 5

**Summary:**

This paper introduces a simple time-series prediction model, MMFNet, based on multi-scale feature fusion using DCT combined with a masking mechanism. The model first divides the input sequence into three different scales, applies DCT transformation to each scale, and obtains frequency domain features at different scales. Then, adaptive filtering is applied to these frequency domain features using a masking operation. A simple linear interpolation layer is used to obtain predictions at each scale, and the final prediction is obtained by applying the inverse DCT and summing these results. The authors validate the model's effectiveness through a series of experiments and demonstrate the importance of the masking mechanism and multi-scale feature fusion to the model's performance through ablation studies.

**Strengths:**

1. The model design appears very simple and achieves good results, surpassing existing mainstream models.
2. The authors validate the effectiveness of multi-scale frequency domain feature extraction and the masking mechanism on the model's performance through a series of visual experiments and ablation studies, providing some explanations for these.
3. The experiments on extending the horizon sufficiently show that the model has excellent long-term sequence prediction capabilities.

**Weaknesses:**

1. The authors did not explain why DCT was chosen rather than other transformations, such as the more common Fourier transform. What's the advantage of DCT? Are there other transformation methods that could be used instead?
2. The authors argued that this is the first model to use multi-scale frequency domain feature fusion, but there is no discussion on the advantage of frequency domain processing compared to time domain multi-scale feature fusion.
3. The paper claimed that multi-scale frequency decomposition can capture short-term fluctuation characteristics, but there is no intuitive result that demonstrates this. Moreover, what's different information captured by the features at different scales?

**Questions:**

1. In the paper,  three specific scales were designed. Why were these particular sizes (2, 24, 720) chosen? Are there other choices for small and medium scales (such as 4, 48)? Is the model's performance sensitive to the choice of these scales?
2. In Figures 2 and 3 of the appendix, the actual output of the mask is shown, but the shape of the mask seems inconsistent with what was described in the text. The segment lengths for fine-scale and intermediate-scale are 2 and 24, respectively, so the mask lengths should correspond to 2 and 24. However, the mask length in the figure seems to be 720. Was there an intermediate step involved? Can the authors clarify this?
3. The mask output seems to exhibit certain low-pass characteristics at fine-scale and intermediate-scale, while showing high-pass characteristics at coarse-scale. Can you further explain this phenomenon?
4. Can you provide some visualizations of the features learned at different scales? For example, the weights of the linear interpolation layer corresponding to different scales and the final output, etc.
5. During the inversion step, the 3 output were first applied with the inverse DCT, then summed up. Is there any reason for this? Because from a theoretical perspective, the inverse DCT is a linear operation, so the sum of the inverse DCT of the 3 outputs is equivalent to the inverse DCT of the sum of the 3 outputs. By swapping the order of the sum and the inverse DCT, the computational complexity can be greatly reduced. Can you explain this design choice?

---

> ### Author Response · Authors · 2024-11-24
>
> ---
>
> ## Comment 1
>
> **Comment:**
> *The authors did not explain why DCT was chosen rather than other transformations, such as the more common Fourier Transform. What's the advantage of DCT? Are there other transformation methods that could be used instead?*
>
> **Response:**
> Thank you for your comment. Compared to the Fourier Transform (FFT or rFFT), the Discrete Cosine Transform (DCT) represents only real-valued frequency components, simplifying the processing pipeline. This reduces the need for complex neural network architectures to perform interpolation, making DCT a more computationally efficient choice for our approach.
>
> ---
>
> ## Comment 2
>
> **Comment:**
> *The authors argued that this is the first model to use multi-scale frequency domain feature fusion, but there is no discussion on the advantage of frequency domain processing compared to time domain multi-scale feature fusion.*
>
> **Response:**
> Thank you for your comment. Frequency domain processing offers distinct advantages over time-domain multi-scale feature fusion, particularly for capturing global and periodic patterns inherent in time series data. While time-domain methods excel at analyzing local and sequential features, they often struggle with efficiently representing periodic and frequency-specific information. Frequency domain processing, such as the Multi-Scale Frequency Transformation used in MMFNet, enables the model to decompose and analyze hierarchical frequency structures, providing a complementary perspective that enhances forecasting accuracy and interpretability.
>
> We have added discussions on these advantages to **Appendix D (Page 24)** of the revised manuscript.
>
> ---
>
> ## Comment 3
>
> **Comment:**
> *The paper claimed that multi-scale frequency decomposition can capture short-term fluctuation characteristics, but there is no intuitive result that demonstrates this. Moreover, what's different information captured by the features at different scales?*
>
> **Response:**
> Thank you for your comment. To demonstrate that multi-scale frequency decomposition captures short-term fluctuation characteristics, we have included additional visualizations in **Appendix C.7 (Page 23)**, showing the contributions of different scales to the final prediction. These visualizations highlight how:
> - **Fine-scale features** focus on localized high-frequency variations, such as abrupt changes or noise, critical for modeling short-term dynamics.
> - **Intermediate-scale features** bridge the gap between fine and coarse scales by capturing medium-range patterns, such as periodic behaviors or sub-seasonal trends.
> - **Coarse-scale features** represent long-term dependencies and global trends, ensuring that the broader structure of the time series is preserved.
>
> By integrating these multi-scale features, MMFNet achieves a balance between capturing localized details and global trends, critical for accurate time series forecasting.
>
> ---
>
> ## Comment 4
>
> **Comment:**
> *In the paper, three specific scales were designed. Why were these particular sizes (2, 24, 720) chosen? Are there other choices for small and medium scales (such as 4, 48)? Is the model's performance sensitive to the choice of these scales?*
>
> **Response:**
> Thank you for your comment. The segment sizes (2, 24, 720) were selected based on their relevance to the characteristics of common datasets:
> - **Segment size of 2**: Captures fine-grained, local patterns.
> - **Segment size of 24**: Corresponds to periodicity observed in many datasets (e.g., daily cycles in ETTh1, ETTh2, Electricity, and Traffic datasets).
> - **Segment size of 720**: Effectively captures global trends and long-term dependencies.
>
> While these scales were effective for our experiments, performance may vary depending on dataset characteristics. For datasets with distinct seasonality, setting the intermediate scale to match the dominant period typically yields better results. Further tuning of scale sizes could optimize performance for specific applications.

---

> ### Author Response · Authors · 2024-11-24
>
> ---
>
> ## Comment 5
>
> **Comment:**
> *In Figures 2 and 3 of the appendix, the actual output of the mask is shown, but the shape of the mask seems inconsistent with what was described in the text. The segment lengths for fine-scale and intermediate-scale are 2 and 24, respectively, so the mask lengths should correspond to 2 and 24. However, the mask length in the figure seems to be 720. Was there an intermediate step involved? Can the authors clarify this?*
>
> **Response:**
> Thank you for your comment. To clarify, the masks shown in **Figures 2 and 3** in the original version are the result of merging the masks for all segments within the sequence. While the mask for each individual segment corresponds to its respective length (e.g., 2 for fine-scale and 24 for intermediate-scale), the merged mask represents the entire sequence, resulting in a length of 720. This merging step was performed for visualization purposes to provide a comprehensive view of the mask's behavior across the entire sequence.
>
> ---
>
> ## Comment 6
>
> **Comment:**
> *The mask output seems to exhibit certain low-pass characteristics at fine-scale and intermediate-scale, while showing high-pass characteristics at coarse-scale. Can you further explain this phenomenon?*
>
> **Response:**
> Thank you for your comment. The characteristics of the mask outputs arise from the hierarchical nature of the multi-scale decomposition and the adaptive frequency masking mechanism:
> - **Fine-scale and intermediate-scale masks (low-pass):** These scales focus on capturing localized patterns and suppress irrelevant high-frequency noise, ensuring critical frequency components for local and medium-range dynamics are retained.
> - **Coarse-scale masks (high-pass):** These scales emphasize residual high-frequency details to complement the broader, global trends represented by low-frequency components.
>
> This behavior reflects the model's ability to adaptively balance the retention and suppression of frequency components across scales, enhancing its capacity to capture both localized and global features. We have added explanations of this phenomenon to **Appendix C.7 (Page 23)** of the revised manuscript.
>
> ---
>
> ## Comment 7
>
> **Comment:**
> *Can you provide some visualizations of the features learned at different scales? For example, the weights of the linear interpolation layer corresponding to different scales and the final output, etc.*
>
> **Response:**
> Thank you for your comment. We have included visualizations of the learned weights and masks across different scales in **Figure 6 (Page 24)** and **Figure 7 (Page 25)**. These figures demonstrate:
> - **Fine-scale features:** Capturing intricate local patterns by focusing on high-frequency components.
> - **Intermediate-scale features:** Bridging fine and coarse details, learning medium-range dynamics.
> - **Coarse-scale features:** Capturing broad trends and low-frequency components, preserving global structure.
>
> These visualizations highlight MMFNet’s ability to perform nuanced multi-scale frequency transformations and adaptive masking, distinguishing it from prior approaches and reinforcing its contributions to time series forecasting.
>
> ---
>
> ## Comment 8
>
> **Comment:**
> *During the inversion step, the 3 outputs were first applied with the inverse DCT, then summed up. Is there any reason for this? Because from a theoretical perspective, the inverse DCT is a linear operation, so the sum of the inverse DCT of the 3 outputs is equivalent to the inverse DCT of the sum of the 3 outputs. By swapping the order of the sum and the inverse DCT, the computational complexity can be greatly reduced. Can you explain this design choice?*
>
> **Response:**
> Thank you for your comment. You are correct that, from a theoretical perspective, swapping the order of the sum and the inverse DCT would reduce computational complexity. However, in practice, we found that this approach negatively impacts prediction performance due to alignment issues of frequency features across different scales. By applying the inverse DCT separately to each scale before summing, we preserve the integrity and alignment of features, ensuring more accurate reconstruction and better overall forecasting performance.

---

> ### Comment · Reviewer_2mgN · 2024-11-25
> **Thanks for your reply**
>
> Thank you for the author's response. Most of my questions have been addressed. After considering the comments from other reviewers and the author's replies, I have decided to keep my original score but increase my confidence from 4 to 5.

---

### Official Review · Reviewer_fzym · 2024-10-31

**Soundness:** 2
**Presentation:** 3
**Contribution:** 3
**Rating:** 5
**Confidence:** 4

**Summary:**

Summary:
This paper introduces MMFNet, a novel model designed to enhance long-term multivariate forecasting by leveraging a multi-scale masked frequency decomposition approach. Experimentation with benchmark datasets shows that MMFNet achieves good performance.

**Strengths:**

Strengths:
1. MMFNet employs multi-scale frequency domain decomposition, exhibiting a certain degree of innovation.
2.MMFNet achieves good performance in a variety of multivariate time series forecasting tasks.
3.The paper is largely understandable.

**Weaknesses:**

Weaknesses:
1.The motivation behind the paper is not clearly articulated. The author mentions: “While single-scale frequency domain decomposition offers a global perspective of time series data in the frequency domain, it lacks the ability to localize specific frequency components within the sequence.” This statement requires further explanation—why is it unable to localize specific frequencies? Similarly, the author needs to further elaborate on why MMFNet can handle non-stationary series (along with detailed examples). These are key points and contributions of the paper.
2.Using only the MSE metric is insufficient. Does the model show similar performance when evaluated using other metrics such as MAE?
3.The visualization work in the paper is not particularly sufficient. The authors should include more comparative figures between Single-Scale Frequency Transformation and Multi-Scale Frequency Transformation, as well as visualizations of the mask effects, and prediction showcases of different models.
4.The authors should compare the models across more dimensions, such as parameter count, training and inference time.
5.The baseline models are not very rich; the current research primarily focuses on analysis at the temporal scale. Recently, there has been some work exploring the variable perspective, yielding good results, such as with the iTransformer and Client models. It is recommended to include these comparisons as well.
6.The authors claim that "Transformer-based architectures have demonstrated exceptional capacity in capturing complex temporal patterns by effectively modeling long-range dependencies through self-attention mechanisms at the cost of heavy computational workload, particularly when facing large-scale time series data, which significantly limits their practicality in real-time applications." This claim is not comprehensive; a more significant limitation of Transformer-based models is their tendency to overfit.

**Questions:**

Questions:
1.What is the length of the historical look-back window used in the experiments reported in Table 1? Were the baselines’ results in Table 1 obtained from other literature or tested by the authors?
2.Can the Multi-Scale Frequency Transformation and Masked Frequency Interpolation methods proposed in the paper be used as plug-and-play components with other time series models?

---

> ### Author Response · Authors · 2024-11-24
>
> ---
>
> ## Comment 1
>
> **Comment:**
> *The motivation behind the paper is not clearly articulated.*
>
> **Response:**
> Thank you for your comment. Time series data are inherently multi-scale and multi-periodic, generated from real-world processes sampled at various scales. Current time-domain multiscale models like TimeMixer are effective at capturing temporal variations but face limitations. They focus mainly on temporal patterns, often missing critical frequency-specific features like harmonics or periodic trends, which are better captured in the frequency domain. Additionally, time-domain methods are sensitive to noise, as they process raw signals directly, allowing noise to propagate across scales and obscure patterns, especially at coarser resolutions. These models also struggle with long-term dependencies, as dividing data into scales can lose the broader context needed for understanding long-range interactions.
>
> MMFNet overcomes these limitations by combining multi-scale frequency domain analysis with dynamic masking. This approach adaptively captures both global and localized patterns, reduces noise, and preserves long-term dependencies.
>
> We have added discussions into the **introduction section (Pages 1–2)** of the revised manuscript to clarify this.
>
> ---
>
> ## Comment 2
>
> **Comment:**
> *The author mentions: “While single-scale frequency domain decomposition offers a global perspective of time series data in the frequency domain, it lacks the ability to localize specific frequency components within the sequence.” This statement requires further explanation—why is it unable to localize specific frequencies? The author needs to further elaborate on why MMFNet can handle non-stationary series (along with detailed examples). These are key points and contributions of the paper.*
>
> **Response:**
> Thank you for your comment. MMFNet performs multi-scale frequency transformations directly in the frequency domain, unlike traditional multi-scale methods that primarily operate in the time domain. This approach captures location-specific information across scales, overcoming the limitations of single-scale frequency methods like FITS, which lose such details. By integrating adaptive masking, MMFNet selectively emphasizes critical frequency components, ensuring a more accurate and robust representation of multi-scale dynamics.
>
> Single-scale frequency decomposition provides a global view of the frequency spectrum but fails to localize specific frequency components within individual segments of the sequence. This limitation is particularly problematic for non-stationary time series, where transient events or high-frequency variations are confined to specific segments. MMFNet addresses this challenge by segmenting the sequence and performing frequency decomposition at multiple scales, effectively capturing both global patterns and localized variations. This ensures that crucial frequency features—such as transient noise or localized cycles—are preserved and utilized for forecasting, as demonstrated in **Figure 2 (Page 14)** and **Figure 3 (Page 15)**.
>
> MMFNet also incorporates adaptive masking, which dynamically filters out noise and irrelevant components at different scales. This enhances robustness, allowing the model to focus on meaningful features. Unlike traditional single-scale methods like FITS, which lack this flexibility, MMFNet excels in handling datasets with high variability or non-stationarity.
>
> ---
>
> ## Comment 3
>
> **Comment:**
> *The visualization work in the paper is not particularly sufficient. The authors should include more comparative figures between Single-Scale Frequency Transformation and Multi-Scale Frequency Transformation, as well as visualizations of the mask effects, and prediction showcases of different models.*
>
> **Response:**
> Thank you for your comment.
>
> 1. **Visualizations of learned weights and masks:**
>    Visualizations at different scales have been added in **Appendix C.7, Figure 6 (Page 24)** and **Figure 7 (Page 25)**. These figures illustrate MMFNet's ability to capture local, medium-range, and global patterns.
>    - Fine-scale visualizations highlight high-frequency components and intricate local details.
>    - Intermediate-scale visualizations bridge fine and coarse patterns, capturing medium-range dynamics.
>    - Coarse-scale visualizations reveal global trends and low-frequency components, demonstrating MMFNet’s adaptability across scales.
>
> 2. **Comparative results between SFT and MFT:**
>    Experiments show that multi-scale combinations consistently outperform Single-Scale Frequency Transformation (SFT). Specifically, the combination \( \text{MFT}(2) + \text{MFT}(360) + \text{MFT}(720) \) achieves the best performance across most horizons and datasets compared with SFT, demonstrating the effectiveness of multi-scale frequency decomposition in capturing hierarchical frequency structures. These results have been added to **Table 8 in Appendix C.3 (Page 20)**.

---

> ### Author Response · Authors · 2024-11-24
>
> ---
>
> ## Comment 4
>
> **Comment:**
> *The authors should compare the models across more dimensions, such as parameter count, training, and inference time.*
>
> **Response:**
> Thank you for your comment. To evaluate computational efficiency, we compared MMFNet with state-of-the-art models on the Electricity dataset for a forecast horizon of 720, assessing parameters, Multiply-Accumulate Operations (MACs), training time (seconds), inference time (milliseconds), and MSE.
> - **MSE:** MMFNet achieves the lowest MSE (0.199), outperforming competitors like PatchTST (0.210) and SparseTSF (0.205).
> - **Efficiency:** MMFNet has only **1.56M parameters** and **499.91M MACs**, significantly lighter than models like FEDformer (**17.98M parameters**).
> - **Inference Time:** MMFNet achieves an inference time of **3.4ms**, second only to DLinear (**1.1ms**).
> - **Training Time:** MMFNet requires **89.2 seconds**, outperforming FEDformer (**238.7 seconds**) while being slightly slower than DLinear (**36.2 seconds**).
>
> We have added these results to **Appendix C.5 (Page 20)** of the revised manuscript.
>
> ---
>
> ## Comment 5
>
> **Comment:**
> *The baseline models are not very rich; the current research primarily focuses on analysis at the temporal scale. Recently, there has been some work exploring the variable perspective, yielding good results, such as with the iTransformer and Client models. It is recommended to include these comparisons as well.*
>
> **Response:**
> Thank you for your comment. To address your concern, we have expanded our experimental evaluation by incorporating three additional datasets—**Health**, **Solar**, and **Exchange**—to assess MMFNet's performance under more diverse and challenging scenarios.
>
> In particular, the **Exchange** dataset, which lacks clear periodicity and seasonality, presents an inherently non-stationary structure with highly unpredictable patterns. Despite these challenges, MMFNet consistently achieves superior prediction performance across these datasets compared to other baseline models, including iTransformer. This demonstrates MMFNet's robustness and adaptability, especially in handling non-stationary time series data effectively.
>
> We have added the results and analysis to **Appendix C.1 (Table 6, Page 17)** of the revised manuscript.
>
> ---
>
> ## Comment 6
>
> **Comment:**
> *Can the Multi-Scale Frequency Transformation and Masked Frequency Interpolation methods proposed in the paper be used as plug-and-play components with other time series models?*
>
> **Response:**
> Thank you for your comment. The Multi-Scale Frequency Transformation (MFT) and Masked Frequency Interpolation (MFI) methods are designed to be lightweight and modular. Since MFT operates as a simple linear layer and MFI consists of learnable neurons, both can seamlessly integrate as plug-and-play components with other time series models, enhancing their capability to capture complex patterns in the frequency domain.

---

> > ### Comment · Reviewer_fzym · 2024-11-30
> > **Response to the authors**
> >
> > Thank you for your response. After careful consideration, although I acknowledge the paper has been enhanced following the revision, I have decided to maintain my original score.
> >
> > 1) Regarding the motivation, the authors have largely reiterated what is already mentioned in the paper, and thus the motivation behind the work still lacks clarity for me.
> >
> > 2) After re-reading the paper and considering the feedback from other reviewers, I believe that the novelty of the work does not appear to be strong enough for ICLR.
> >
> > 3) The authors have not fully addressed all the experiments requested by the reviewers, such as the evaluation metrics and baseline, and the experimental validation could be more comprehensive.

---

### Official Review · Reviewer_dkg8 · 2024-11-01

**Soundness:** 2
**Presentation:** 2
**Contribution:** 1
**Rating:** 3
**Confidence:** 4

**Summary:**

This paper presents MMFNet, a multi-scale frequency masking model for long-term time series forecasting. Unlike most existing models that operate in the time domain, MMFNet follows the light-weight model, FITS (ICLR’24), to learn and capture the dynamic variations in the frequency domain. Compared with FITS, MMFNet has made the following improvements: (1) the multi-scale frequency decomposition to effectively capture both local and global patterns in data; (2) the learnable frequency mask to focus on the most informative frequency components. MMFNet achieves competitive performance to SOTA baselines on several widely-used long-term time series forecasting (LTSF) datasets.

**Strengths:**

- The paper is well-organized and highly readable, and the figures makes it easy to understand the model designs.
- Authors provide ablation analysis to demonstrate the effectiveness of the proposed techniques, i.e,  the multi-scale frequency decomposition and learnable masking.
- I especially appreciated the exploration of  frequency domain for time series modeling, and the experiments show its promising performance.

**Weaknesses:**

- MMFNet seems to be an incremental improvement over FITS, and it appears that the motivation and claim are not well-supported. Specifically, MMFNet may not adequately address the“non-stationary characteristics that SFT may "overlook". Since learnable masks are trained and shared across the whole time series dataset, different time series will be masked out in the same way (via the shared masks).  Therefore, the learnable mask is not so “adaptive”.

- The paper lacks some details. For methodology, it is unclear how to interpolate $X_{DCT}$ of various-length segments and merge them together, some formulas are needed. For experiment settings, the segment num of different temporal scales and the look-back window lengths for various forecasting horizons are not mentioned. I am especially curious about input window length for ultra-long-term forecasting.

- The experimental evaluation seems insufficient. For ablation study, in section 4.3, the impact of different segment num combinations on MMFT should be explored; in section 4.4, the results comparing “Mask” vs. “low pass filter” should be provided to support the paper’s claim. Additionally, I would greatly appreciate it if more comprehensive comparisons could be included in the paper. For examples, the results on different look-back window lengths and visualization of prediction results.

**Questions:**

More discussions:
- FITS can handle the anomaly detection task, does MMFNet also have such capabilities? Since the MMFNet adopts a masking and interpolation approach, it seems more suitable for reconstruction tasks?
- Is the number of model parameters large? From my understanding, each segment has its own learnable mask and linear layer. Consequently, it appears that increasing the number of segments could lead to a significant expansion in the model's parameter count.

---

> ### Author Response · Authors · 2024-11-24
>
> ---
>
> ## Comment 1
>
> **Comment:**
> *MMFNet seems to be an incremental improvement over FITS, and it appears that the motivation and claim are not well-supported. Specifically, MMFNet may not adequately address the "non-stationary characteristics that SFT may overlook." Since learnable masks are trained and shared across the whole time series dataset, different time series will be masked out in the same way (via the shared masks). Therefore, the learnable mask is not so "adaptive."*
>
> **Response:**
> Thank you for your comment. While FITS applies frequency interpolation in the frequency domain, it relies on single-scale frequency decomposition, which captures the overall frequency spectrum but lacks the ability to localize specific frequency components within individual segments. This limitation is particularly problematic for non-stationary time series, where frequency characteristics evolve over time. High-frequency noise or transient events often appear in specific segments but are blurred in single-scale analysis, which produces similar spectra across segments and loses location-specific details, as shown in **Figure 2 (Page 14)**. This hinders accurate forecasting, particularly in scenarios requiring the identification of subtle temporal and spectral variations.
>
> To address this issue, MMFNet introduces Multi-scale Frequency Transformation (MFT) to enable localized frequency domain analysis. By performing frequency decomposition at multiple scales, MMFNet captures both global patterns and localized variations, preserving critical frequency features such as transient events or localized cycles. As shown in **Figure 3 (Page 15)**, MFT ensures that location-specific frequency details are retained across segments, effectively modeling hierarchical and nested structures. Unlike single-scale methods like FITS, MFT leverages a dynamic masking mechanism to adaptively filter out noise, ensuring robustness and focus on meaningful components. This combination of multi-scale analysis and adaptive filtering allows MMFNet to handle non-stationary data more effectively, offering superior performance compared to single-scale methods. Further details on our contributions are highlighted in **Appendix A (Pages 14–15)**.
>
> ---
>
> ## Comment 2
>
> **Comment:**
> *The experimental evaluation seems insufficient. For the ablation study, in Section 4.3, the impact of different segment combinations on MMFT should be explored; in Section 4.4, the results comparing "Mask" vs. "low pass filter" should be provided to support the paper’s claim. Additionally, I would greatly appreciate it if more comprehensive comparisons could be included in the paper. For example, the results on different look-back window lengths and visualization of prediction results.*
>
> **Response:**
> To evaluate the prediction performance of MMFNet across different segment combinations, we conducted experiments. **Table 8 (Page 20)** presents the MSE values for various configurations of Multi-scale Frequency Transformation (MFT) applied to the Solar and Exchange datasets. MFT involves performing masked frequency transformation with fragmentation at different scales to capture fine-grained, intermediate, and coarse-grained frequency characteristics. Segment lengths of 2, 360, and 720 are used to represent fine-scale, intermediate-scale, and coarse-scale MFT, respectively.
>
> For each dataset, the results are reported across four forecasting horizons: 96, 192, 336, and 720. The configurations include individual MFT scales, pairwise combinations (e.g., \( \text{MFT}(360) + \text{MFT}(720) \)), and a comprehensive multi-scale combination (\( \text{MFT}(2) + \text{MFT}(360) + \text{MFT}(720) \)).
>
> The results show that combining multiple scales generally improves forecasting performance compared to single-scale MFT. Notably, the comprehensive combination of \( \text{MFT}(2) + \text{MFT}(360) + \text{MFT}(720) \) achieves the best performance across most horizons and datasets. This demonstrates the effectiveness of multi-scale frequency decomposition in capturing hierarchical and nested frequency structures in time series data. The lower MSE values highlight MMFNet's ability to leverage multi-scale information for more accurate predictions.
>
> New experimental results have been added to **Table 8 in Appendix C.3 (Page 20)** of the revised manuscript.
>
> ---

---

> ### Author Response · Authors · 2024-11-24
>
> ## Comment 3
>
> **Comment:**
> *FITS can handle the anomaly detection task. Does MMFNet also have such capabilities? Since MMFNet adopts a masking and interpolation approach, it seems more suitable for reconstruction tasks.*
>
> **Response:**
> Thank you for your comment. To assess the anomaly detection performance of MMFNet, we conducted experiments on three datasets: **SMD**, **MSL**, and **SMAP**, and compared its results with those of other models. MMFNet achieves remarkable performance, securing the highest F1-scores on **MSL (90.74)** and **SMD (71.27)**, while delivering competitive results for **SMAP** with an F1-score of **67.42**. Furthermore, MMFNet attains the highest average Precision (Avg P) at **86.22%**, showcasing its robustness and precision across datasets.
>
> These findings emphasize MMFNet's ability to effectively balance Precision and Recall, establishing it as one of the most reliable models for anomaly detection tasks across diverse datasets. Its superior performance and high average precision underline MMFNet's consistency and robustness in handling various anomaly detection scenarios. New experimental results have been added to **Appendix C.2 (Table 7, Page 19)** of the revised manuscript.
>
> ---
>
> ## Comment 4
>
> **Comment:**
> *Is the number of model parameters large? From my understanding, each segment has its own learnable mask and linear layer. Consequently, it appears that increasing the number of segments could lead to a significant expansion in the model's parameter count.*
>
> **Response:**
> Thank you for your comment. We conducted efficiency experiments on the Electricity dataset. MMFNet achieves the lowest MSE of **0.199**, highlighting its superior prediction accuracy compared to competing models such as PatchTST (**0.201**) and SparseTSF (**0.205**). Additionally, it excels in computational efficiency, with only **1.56M parameters** and **499.91M MACs**—significantly lower than larger models like FEDformer (**17.98M parameters**). MMFNet’s inference time of **3.4ms** makes it one of the fastest models, second only to DLinear at **1.1ms**. Its training time of **89.2 seconds** is highly competitive, outperforming models like FEDformer (**238.7 seconds**) while being slightly slower than DLinear (**36.2 seconds**). We have added the results to **Table 10 in Appendix C.5 (Page 21)** of the revised manuscript.
>
> ---
>
> ## Comment 5
>
> **Comment:**
> *The paper lacks some details. For methodology, it is unclear how to interpolate XDCT of various-length segments and merge them together—some formulas are needed. For experiment settings, the segment number of different temporal scales and the look-back window lengths for various forecasting horizons are not mentioned. I am especially curious about the input window length for ultra-long-term forecasting.*
>
> **Response:**
> Thank you for your comment. For each scale, we merge the Discrete Cosine Transform (DCT) spectrum of the segments and apply a linear layer to reconstruct the prediction spectrum. Finally, the reconstructed prediction spectra from all scales are merged to generate the final output. Regarding the experimental settings, the input window length for ultra-long-term forecasting is set to **1440**.

---

> > ### Comment · Reviewer_dkg8 · 2024-11-26
> >
> > Thank you for the authors' response, and I appreciate their efforts in adding more experimental evaluation. My concerns regarding the paper's motivation persist, as these are key points.
> >
> > **About the paper motivation:**
> >
> > I agree that MFT is a more flexible operation than single-scale frequency decomposition because MFT splits the entire series into different-scale segments and retains various location-specific frequency components across these segments. However, the authors' argument that MMFNet can address the non-stationarity of time series remains unconvincing. In my understanding, the masks are learned dynamically on each dataset and then fixed for testing. However, to alleviate/solve the non-stationarity, the masks should adapt to statistical properties of different input sequences.
> >
> > Additionally, I noticed that the authors made some changes to the paper's motivation in the introduction, and some descriptions about FITS were removed. I have two suggestions: First, the authors should be cautious about the paper's motivation, especially regarding the non-stationarity aspect. Second, if the paper is inspired by FITS, the authors should still mention more details of FITS in the introduction and reasonably propose the research gap.
> >
> >
> > **About the anomaly detection task:**
> >
> > It would be beneficial to include FITS as a baseline since FITS is the "predecessor" of MMFNet. However, considering that this paper focuses on time series forecasting, it is acceptable to provide brief experimental results to demonstrate MMFNet's effectiveness.
> >
> > **About some details:**
> >
> > " For each scale, we merge the Discrete Cosine Transform (DCT) spectrum of the segments and apply a linear layer to reconstruct the prediction spectrum".  Here, "merge" could mean "concatenate" (?), but it's not explicitly clear. It would be better to specify the operation for clarity.
> >
> > "Finally, the reconstructed prediction spectra from all scales are merged to generate the final output".
> > Here, "merged" could mean "added" (?),  but again, it's not clear. Specifying the operation (e.g., summed, averaged, etc.) would be more precise.
> >
> > Additionally, regarding the "segment number of different temporal scales and the look-back window lengths," are they consistent across different datasets, or do they vary?

---

> > > ### Author Response · Authors · 2024-11-26
> > >
> > > **Dear Reviewer,**
> > >
> > > Thank you for your thoughtful comments and detailed feedback. We appreciate the time and effort you have taken to evaluate our work. Below are our responses to your concerns:
> > >
> > > ---
> > >
> > > >About the paper motivation:
> > >
> > > We have added additional details about FITS in the updated version of the introduction, which can be found in Appendix G (Page 28). We also acknowledge your concern about the non-stationarity aspect and will be more cautious in addressing it. When we refer to non-stationarity in the paper, we specifically mean periodic time series that exhibit certain non-stationary characteristics.
> > >
> > > ---
> > >
> > > >About the anomaly detection task:
> > >
> > > We have included the experimental results of FITS on the anomaly detection task in Appendix C.2 (Table 7, Page 19). The results demonstrate that MMFNet outperforms FITS in this task, further showcasing the effectiveness of our approach.
> > >
> > > ---
> > >
> > > >About some details:
> > >
> > >  **Merging information across scales:**
> > >    We merge information from different scales by averaging. Additional details about this process have been included in Section 3.4 (Pages 5–6).
> > >
> > >  **Segment number of different temporal scales:**
> > >    - The fine scale is set to 2,
> > >    - The coarse scale is set to 360,
> > >    - The intermediate scale typically corresponds to the period of the time series.
> > >
> > > **Look-back window lengths:**
> > >    - For most datasets, we use a look-back window of 1440.
> > >    - However, for ETTh1 and ETTh2, we apply a look-back window of 720 or 512 to optimize prediction performance.
> > >
> > >
> > > ---

---

> > > > ### Comment · Reviewer_dkg8 · 2024-12-02
> > > >
> > > > Thank you for your response. After carefully reviewing the comments and the revised paper, I have decided to maintain my original score. My reservations regarding the paper's motivation remain, as the motivation has been restated rather than substantially developed. Additionally, while the paper offers an incremental improvement over FITS, the novelty and contributions do not seem to be enough for ICLR.

---

### Official Review · Reviewer_bYj9 · 2024-11-03

**Soundness:** 3
**Presentation:** 3
**Contribution:** 2
**Rating:** 3
**Confidence:** 3

**Summary:**

MMFNet is a predictive model designed for non-stationary time series, addressing the limitations of Single-scale Frequency Decomposition methods like FITS, which struggle to capture non-stationary patterns and may miss short-term fluctuations. To overcome these limitations, MMFNet introduces a Multi-scale Frequency Decomposition approach. Additionally, it incorporates a learnable mask that selectively filters out irrelevant frequency components at each scale, thereby enhancing prediction accuracy and effectively capturing both short-term variations and long-term trends.

**Strengths:**

1. Clear Identification of Limitations: The paper effectively identifies the limitations of existing methods like FITS, particularly in handling non-stationary data and capturing short-term fluctuations. This acknowledgment establishes a strong rationale and relevance for the proposed research.
2. Well-Organized Presentation: The methodology is well-organized and explained step-by-step, making the approach accessible and easier for readers to understand.

**Weaknesses:**

1. Unclear Motivation: The paper aims to analyze time series data across multiple scales, an approach that has also been explored by methods like TimeMixer, PITS, and FITS. However, it lacks a clear motivation for why this particular approach is necessary and does not sufficiently explain how MMFNet differs from or improves upon these existing methods. It would help if the authors more explicitly justified their approach by highlighting the specific advantages or unique aspects of MMFNet that set it apart from similar multi-scale methods.

2. Limited Novelty: The proposed method feels more like a variation on established approaches rather than a fundamentally new concept. Both multi-scale analysis and frequency masking techniques have been explored previously in time series research, making it challenging to see where MMFNet offers substantial innovation. The paper could benefit from a clearer articulation of its novel aspects, such as whether MMFNet provides specific advancements in frequency decomposition or interpretability that distinguish it from other multi-scale approaches.

3. Insufficient Comparative Experiments with Existing Models: The paper lacks enough experimental comparisons with other prominent models (e.g., TimeMixer) to effectively demonstrate any performance advantages or unique capabilities of MMFNet. Direct comparisons on standard datasets with similar metrics could better showcase the proposed model’s strengths and allow readers to more easily assess its advantages over existing approaches.

4. Limited Persuasiveness Due to Lack of Novelty: Given the number of studies on multi-scale time series analysis, the paper does not provide sufficient differentiation to persuade readers of MMFNet's unique contributions. Expanding the explanation of MMFNet’s specific innovations and benefits could strengthen its positioning as a valuable addition to the existing research.

**Questions:**

1. How does the proposed Multi-scale Frequency Decomposition specifically address non-stationarity?
Further explanation of how multi-scale decomposition effectively handles non-stationary characteristics would help clarify this point.

2. What are the limitations of the multi-scale approach, and are there any suggested improvements?
Every multi-scale approach may have limitations, such as increased noise or computational complexity. It would be valuable to discuss any potential drawbacks of this method and ways to mitigate them.

3. How does the proposed method compare in terms of computational cost or time complexity to existing models?
Providing a comparison of computational efficiency would help readers understand the practicality of the proposed model.

4. Are there experimental results on the National Illness dataset?
Results on such datasets would help demonstrate the relevance and applicability of the method, particularly for health-related time series data.

**Details Of Ethics Concerns:**

.

---

> ### Author Response · Authors · 2024-11-24
>
> ## Comment 1
>
> **Comment:**
> The paper aims to analyze time series data across multiple scales, an approach that has also been explored by methods like TimeMixer, PITS, and FITS. However, it lacks a clear motivation for why this particular approach is necessary and does not sufficiently explain how MMFNet differs from or improves upon these existing methods.
>
> **Response:**
> Thank you for your comment. Time series data are inherently multi-scale and multi-periodic, generated from real-world processes sampled at various scales. For example, daily data capture hourly fluctuations, while yearly data reflect long-term trends and seasonal cycles. This nature makes model design challenging, as each scale emphasizes different temporal dynamics. Centered Kernel Alignment (CKA) analysis shows that diverse representations across layers help manage variations across scales and periodicities, particularly for tasks involving irregular patterns.
>
> Current time-domain multiscale models like TimeMixer are effective at capturing temporal variations but face limitations. They focus mainly on temporal patterns, often missing critical frequency-specific features like harmonics or periodic trends, which are better captured in the frequency domain. Additionally, time-domain methods are sensitive to noise, as they process raw signals directly, allowing noise to propagate across scales and obscure patterns, especially at coarser resolutions. These models also struggle with long-term dependencies, as dividing data into scales can lose the broader context needed for understanding long-range interactions.
>
> MMFNet overcomes these limitations by combining multi-scale frequency domain analysis with dynamic masking. This approach adaptively captures both global and localized patterns, reduces noise, and preserves long-term dependencies. We believe MMFNet’s unique design addresses key gaps in current methods and offers significant improvements in handling multi-scale and multi-periodic time series data.
>
> We have added discussions into the introduction section (Pages 1–2) of the revised manuscript.
>
> ---
>
> ## Comment 2
>
> **Comment:**
> The proposed method feels more like a variation on established approaches rather than a fundamentally new concept. Both multi-scale analysis and frequency masking techniques have been explored previously in time series research, making it challenging to see where MMFNet offers substantial innovation.
>
> **Response:**
> Thank you for your comment. While multi-scale analysis and frequency masking have been explored in prior research, MMFNet introduces distinct innovations that address critical limitations of existing methods.
>
> MMFNet performs multi-scale frequency transformations directly in the frequency domain, unlike traditional multi-scale methods that primarily operate in the time domain. This approach captures location-specific information across scales, overcoming the limitations of single-scale frequency methods like FITS, which lose such details. By integrating adaptive masking, MMFNet selectively emphasizes critical frequency components, ensuring a more accurate and robust representation of multi-scale dynamics.
>
> Single-scale frequency decomposition provides a global view of the frequency spectrum but fails to localize specific frequency components within individual segments of the sequence. This limitation is particularly problematic for non-stationary time series, where transient events or high-frequency variations are confined to specific segments. MMFNet addresses this challenge by segmenting the sequence and performing frequency decomposition at multiple scales, effectively capturing both global patterns and localized variations. This ensures that crucial frequency features—such as transient noise or localized cycles—are preserved and utilized for forecasting, as demonstrated in **Figure 2 (Page 14)** and **Figure 3 (Page 15)**.
>
> MMFNet also incorporates adaptive masking, which dynamically filters out noise and irrelevant components at different scales. This enhances robustness, allowing the model to focus on meaningful features. Unlike traditional single-scale methods like FITS, which lack this flexibility, MMFNet excels in handling datasets with high variability or non-stationarity.
>
> Visualizations in **Figures 6 (Page 24)** and **7 (Page 25)** further illustrate the contributions of MMFNet. At the fine scale, the model captures local high-frequency patterns; at the intermediate scale, it balances local and global dynamics; and at the coarse scale, it identifies broad trends and long-term dependencies. These visualizations highlight how MMFNet effectively balances local and global representations. MMFNet’s novel combination of multi-scale frequency transformation and adaptive masking delivers superior performance and robustness in handling complex, dynamic time series data.
>
> ---

---

> ### Author Response · Authors · 2024-11-24
>
> ## Comment 3
>
> **Comment:**
> *Insufficient Comparative Experiments with Existing Models: The paper lacks enough experimental comparisons with other prominent models (e.g., TimeMixer) to effectively demonstrate any performance advantages or unique capabilities of MMFNet.*
>
> **Response:**
> Thank you for your comment. To address this issue, we have expanded our experimental evaluation by incorporating three additional datasets—**Health**, **Solar**, and **Exchange**—to assess MMFNet's performance under more diverse and challenging scenarios.
>
> In particular, the **Exchange** dataset, which lacks clear periodicity and seasonality, presents an inherently non-stationary structure with highly unpredictable patterns. Despite these challenges, MMFNet consistently achieves superior prediction performance across these datasets compared to other baseline models, including TimeMixer. This demonstrates MMFNet's robustness and adaptability, especially in handling non-stationary time series data effectively.
>
> Detailed results and analysis are provided in **Appendix C.1 (Table 6, Page 18)**, showcasing MMFNet's performance and highlighting its ability to excel under such conditions.
>
> ---
>
> ## Comment 4
>
> **Comment:**
> *How does the proposed Multi-scale Frequency Decomposition specifically address non-stationarity?*
>
> **Response:**
> Thank you for your comment. MMFNet addresses non-stationarity by performing multi-scale frequency transformations directly in the frequency domain. This approach enables the model to capture location-specific information across scales while employing adaptive masking to emphasize critical frequency components, making it highly effective in handling non-stationary patterns.
>
> To further clarify the contributions of the Multi-scale Frequency Transformation (MFT) and adaptive masking mechanisms, we have provided visualizations in **Figure 6 (Page 24)** and **Figure 7 (Page 25)**. These figures illustrate how MMFNet captures patterns at different scales, from intricate local details and medium-range dynamics to broad trends and long-term dependencies. These visualizations underscore MMFNet’s ability to perform nuanced multi-scale frequency transformations and adaptive masking, setting it apart from existing methods and enhancing its robustness in time series forecasting.
>
> ---
>
> ## Comment 5
>
> **Comment:**
> *Are there experimental results on the National Illness dataset?*
>
> **Response:**
> Thank you for your comment. We have added results on the **National Illness** dataset. Experiments show MMFNet delivers top-2 prediction performance on this dataset. The new experimental results have been added to **Table 6 in Appendix C.1 (Page 18)** of the revised manuscript.
>
> ---
>
> ## Comment 6
>
> **Comment:**
> *How does the proposed method compare in terms of computational cost or time complexity to existing models?*
>
> **Response:**
> Thank you for your comment.
>
> To evaluate computational efficiency, we compare MMFNet with state-of-the-art models on the Electricity dataset for a forecast horizon of 720, assessing parameters, Multiply-Accumulate Operations (MACs), training time (seconds), inference time (milliseconds), and MSE. MMFNet achieves the lowest MSE (0.199), outperforming competitors like PatchTST (0.210) and SparseTSF (0.205). It maintains a favorable computational profile with **1.56M parameters** and **499.91M MACs**, significantly lighter than models like FEDformer (17.98M parameters). With an inference time of **3.4ms** and training time of **89.2 seconds**, MMFNet strikes a balance between efficiency and accuracy, making it a strong candidate for time series forecasting tasks requiring both precision and computational feasibility.
>
> The results are included in **Appendix C.5 (Page 21)**.

---

### Official Review · Reviewer_MCJL · 2024-11-04

**Soundness:** 2
**Presentation:** 3
**Contribution:** 2
**Rating:** 3
**Confidence:** 4

**Summary:**

This paper introduces MMFNet, a model aimed at enhancing long-term time series forecasting (LTSF) by employing a multi-scale frequency decomposition technique. Addressing limitations of existing models that assume stationarity and filter out high-frequency components—thus potentially overlooking short-term fluctuations—MMFNet captures both short-term variations and long-term trends by decomposing time series data into fine, intermediate, and coarse segments. These segments are selectively filtered through a learnable mask to focus on relevant frequencies.

**Strengths:**

- The paper is well-organized, with a logical flow from the introduction to the proposed method and experiments
- The motivation section highlights the limitations of current LTSF models in capturing both short-term fluctuations and long-term trends, emphasizing the need for a model that can balance these aspects. By framing MMFNet as a solution to this gap, the paper sets a compelling context for its multi-scale approach.
- The experiments cover a range of popular LTSF datasets (e.g. ETT, Weather, Traffic, and Electricity), and inclusion of both high-channel and low-channel datasets helps validate the model’s versatility in different settings.

**Weaknesses:**

Despite the performance gains across multiple datasets, the motivation is not fully substantiated:

- The rationale for multi-scale frequency decomposition lacks empirical grounding, as the manuscript does not convincingly demonstrate why a single-scale approach falls short. While the authors reference assumptions of stationarity, they do not provide empirical evidence or citations that highlight the limitations of single-scale models in specific LTSF scenarios.

Additionally, several design choices require further explanation:

- The learnable mask's role in frequency filtering is not well-clarified in terms of its operational benefits over traditional fixed-frequency filters. While described as "self-adaptive," the paper does not detail how the mask parameters are optimized or discuss their consistent effectiveness across datasets. For instance, the appendix includes some masking results, but without analysis on how these masks contribute to the model’s forecasting accuracy.

Improving the experimental design to highlight robustness in non-stationary settings would enhance the study's validity:

- Although MMFNet achieves moderate performance improvements on benchmark datasets, its robustness to non-stationary data remains unclear. The experiments do not include clearly non-stationary datasets or demonstrate superior performance under such conditions. The chosen datasets, like ETTh1 and Electricity, mainly display seasonal patterns, making it uncertain if MMFNet is truly resilient to non-stationary series.

Lastly, computational analysis and comparisons to related works [1][2] would be beneficial:

- The multi-scale decomposition and frequency masking likely introduce computational overhead, but the paper does not quantify these costs or evaluate MMFNet’s efficiency in practical applications. A detailed comparison of MMFNet’s computational complexity against simpler models would clarify the trade-offs between overhead and performance. Additionally, recent relevant baselines, such as [2], have not been addressed in the manuscript.

[1] FITS: Modeling Time Series with 10k Parameters (ICLR 2024)

[2] DAM: TOWARDS A FOUNDATION MODEL FOR TIME SERIES FORECASTING (ICLR 2024)

**Questions:**

In conclusion, while the paper proposes a novel approach to long-term time series forecasting through multi-scale frequency decomposition, the necessity of this complex approach is not fully justified. Although the motivation to capture both short-term fluctuations and long-term trends is valid, the rationale behind key design choices, such as the adaptive masking mechanism, is underexplained. Furthermore, the model’s empirical gains over existing baselines are modest and inconsistent across datasets, reducing the impact of the experimental findings. Nonetheless, the paper is well-organized, and the architecture shows potential, especially if the authors can clarify the interpretability of the frequency masking mechanism and strengthen the model's practical applicability through more targeted benchmarks.

---

> ### Author Response · Authors · 2024-11-24
>
> ### Comment 1
>
> **Comment:**
> *The rationale for multi-scale frequency decomposition lacks empirical grounding, as the manuscript does not convincingly demonstrate why a single-scale approach falls short. While the authors reference assumptions of stationarity, they do not provide empirical evidence or citations that highlight the limitations of single-scale models in specific LTSF scenarios.*
>
> **Response:**
> Thank you for your comment. Single-scale frequency decomposition provides a global view of the time series by analyzing the overall frequency spectrum but struggles with non-stationary data where frequency characteristics change over time. It cannot localize specific frequency components to particular segments, which is critical for capturing transient events or high-frequency variations.
>
> As shown in **Figure 2 (Page 14)**, this limitation leads to similar spectra across segments, losing vital location-specific details and reducing accuracy, especially in complex multivariate forecasting. To address this, we propose MMFNet with its core component, Multi-scale Frequency Transformation, as shown in **Figure 3 (Page 15)**. MMFNet performs frequency decomposition at multiple scales, capturing both global patterns and localized variations. Unlike single-scale methods like FiTS, MMFNet preserves location-specific frequency details, enabling the model to effectively capture hierarchical and nested structures. Furthermore, MMFNet uses dynamic masking to adaptively filter out noise, ensuring robustness to irrelevant components and variability in the data.
>
> By combining multi-scale decomposition and adaptive noise filtering, MMFNet bridges the gap between global and localized analysis, delivering superior performance compared to single-scale methods. This makes MMFNet particularly effective for forecasting tasks involving complex, dynamic time series.
>
> ---
>
> ### Comment 2
>
> **Comment:**
> *The learnable mask's role in frequency filtering is not well-clarified in terms of its operational benefits over traditional fixed-frequency filters.*
>
> **Response:**
> Thank you for your comment. To address this issue, we have provided visualizations of the weights and masks learned at different scales in **Figure 6 (Page 24)** and **Figure 7 (Page 25)**.
>
> - The fine-scale weight plot (top left) shows the model’s weight assignments at the highest resolution (segment length = 2), focusing on local patterns within the time series and capturing detailed variations.
> - The intermediate-scale weight plot (middle left) illustrates weights learned at a moderate resolution (segment length = 360), effectively bridging the gap between localized details and broader trends.
> - The coarse-scale weight plot (bottom left) highlights the weights at the lowest resolution (segment length = 1440), emphasizing global trends and low-frequency components across the entire sequence.
>
> These plots demonstrate how the learnable masks dynamically adapt to different temporal resolutions, offering a clear advantage over traditional fixed-frequency filters. Unlike fixed filters, the learnable masks tailor frequency filtering to the specific characteristics of each scale, enabling the model to capture both fine-grained details and global patterns more effectively. This adaptability enhances forecasting accuracy and better aligns with the diverse temporal dynamics of real-world time series. We hope this explanation, along with the provided visualizations, clarifies the operational benefits of the learnable masks.
>
> ---

---

> ### Author Response · Authors · 2024-11-24
>
> ### Comment 3
>
> **Comment:**
> *Although MMFNet achieves moderate performance improvements on benchmark datasets, its robustness to non-stationary data remains unclear. The experiments do not include clearly non-stationary datasets or demonstrate superior performance under such conditions. The chosen datasets, like ETTh1 and Electricity, mainly display seasonal patterns, making it uncertain if MMFNet is truly resilient to non-stationary series.*
>
> **Response:**
> Thank you for your comment. To address this issue, we have incorporated three additional datasets (i.e., **Health**, **Solar**, and **Exchange**) to further evaluate MMFNet's performance under more challenging scenarios. In particular, the Exchange dataset lacks clear periodicity and seasonality, presenting an inherently non-stationary structure with unpredictable patterns.
>
> Despite these challenges, MMFNet consistently achieves superior prediction performance, outperforming other baseline models across these datasets. This demonstrates its robustness and effectiveness in handling non-stationary time series data. Detailed results and analysis have been provided in **Appendix C.1 (Table 6, Page 18)** to highlight MMFNet's ability to adapt to and excel in such conditions.
>
>
> ---
>
> ### Comment 4
>
> **Comment:**
> *Lastly, computational analysis and comparisons to related works [1][2] would be beneficial.*
>
> **Response:**
> Thank you for your comment. MMFNet achieves the lowest MSE of **0.199**, highlighting its superior prediction accuracy compared to competing models such as PatchTST (0.201) and SparseTSF (0.205). Additionally, it excels in computational efficiency, with only **1.56M parameters** and **499.91M MACs**—significantly lower than larger models like FEDformer (17.98M parameters).
>
> MMFNet's inference time of **3.4ms** makes it one of the fastest models, second only to DLinear at **1.1ms**. Its training time of **89.2 seconds** is highly competitive, outperforming models like FEDformer (**238.7 seconds**) while being slightly slower than DLinear (**36.2 seconds**). The detailed results are available in **Table 10, Appendix C.5 (Page 21)**.

---

> > ### Comment · Reviewer_MCJL · 2024-12-03
> >
> > Thank you for your response. After thoroughly reviewing the comments and the revised paper, I have decided to maintain my initial score. While the authors have reclarified some minor weaknesses I’ve pointed out, my concerns regarding the paper's motivation remain.

---

### Meta-Review · Area_Chair_Nyah · 2024-12-16

**Metareview:**

The authors propose to capture multi-scale patterns in the frequency domain for time series forecasting. One of their motivation is existing methods are overlooking the importance of the harmonic, periodic patterns in the frequency domain. As I know, however, this claim is not true. One example is TimeMixer [Wang et al., ICLR 2024] which uses the multi-scale frequency decomposition. In this regard, this paper's motivation fails to reflect recent research outcomes for time series forecasting. I recommend that the authors carefully revise the motivation and the model design. Reviewers also initially raised issues regarding the motivation and the novelty of the paper. Even after the rebuttal discussion, they were not all addressed. We strongly recommend that the authors significantly improve the quality of the paper before resubmission.

**Additional Comments On Reviewer Discussion:**

The authors tried to justify their work in terms of the motivation, the novelty, and the efficacy in various time series datasets. However, reviewers mostly did not accept the justification and stayed on their initial score.

---

### Decision · Program_Chairs · 2025-01-22

Reject